# Congenital hearing impairment associated with peripheral cochlear nerve dysmyelination in glycosylation-deficient muscular dystrophy

**Shigefumi Morioka[1,2], Hirofumi Sakaguchi[2], Hiroaki Mohri[1], Mariko Taniguchi-Ikeda[3,4], Motoi Kanagawa[3¤], Toshiaki Suzuki[1], Yuko Miyagoe-Suzuki[5], Tatsushi Toda[3,6], Naoaki Saito[1], Takehiko Ueyama[1]** *

1 Laboratory of Molecular Pharmacology, Biosignal Research Center, Kobe University, Kobe, Japan,
2 Department of Otolaryngology-Head and Neck Surgery, Kyoto Prefectural University of Medicine, Kyoto, Japan, 3 Division of Molecular Brain Science, Kobe University Graduate School of Medicine, Kobe, Japan, 4 Department of Clinical Genetics, Fujita Health University Hospital, Toyoake, Japan, 5 Department of Molecular Therapy, National Institute of Neuroscience, National Center of Neurology and Psychiatry, Tokyo, Japan, 6 Department of Neurology, Graduate School of Medicine, The University of Tokyo, Tokyo, Japan

¤ Current address: Department of Molecular Cardiovascular Biology and Pharmacology, Ehime University Graduate School of Medicine, Shitsukawa, Japan
* tueyama@kobe-u.ac.jp

## Abstract

Hearing loss (HL) is one of the most common sensory impairments and etiologically and genetically heterogeneous disorders in humans. Muscular dystrophies (MDs) are neuromuscular disorders characterized by progressive degeneration of skeletal muscle accompanied by non-muscular symptoms. Aberrant glycosylation of α-dystroglycan causes at least eighteen subtypes of MD, now categorized as MD-dystroglycanopathy (MD-DG), with a wide spectrum of non-muscular symptoms. Despite a growing number of MD-DG subtypes and increasing evidence regarding their molecular pathogeneses, no comprehensive study has investigated sensorineural HL (SNHL) in MD-DG. Here, we found that two mouse models of MD-DG, *Large^{myd/myd}* and *POMGnT1*-KO mice, exhibited congenital, non-progressive, and mild-to-moderate SNHL in auditory brainstem response (ABR) accompanied by extended latency of wave I. Profoundly abnormal myelination was found at the peripheral segment of the cochlear nerve, which is rich in the glycosylated α-dystroglycan–laminin complex and demarcated by "the glial dome." In addition, patients with Fukuyama congenital MD, a type of MD-DG, also had latent SNHL with extended latency of wave I in ABR. Collectively, these findings indicate that hearing impairment associated with impaired Schwann cell-mediated myelination at the peripheral segment of the cochlear nerve is a notable symptom of MD-DG.

**Data Availability Statement:** All relevant data are within the manuscript and its Supporting Information files.

**Funding:** This work was supported by JSPS KAKENHI (#17H04042 and #19K22472 to TU, and #18K09383 to HS), the AMED (JP19ek0109398 to TU), the Uehara Foundation (# 201320273 to TU), the Hyogo Science and Technology Association (#30075 to TU), the Naito Foundation (to TU), and the joint research program of the Biosignal Research Center, Kobe University (#281005, #291004 and #301004 to HS). The funders had no role in study design, data collection and analysis, decision to publish, or preparation of the manuscript.

**Competing interests:** The authors have declared that no competing interests exist.

## Author summary

Hearing loss (HL) is one of the most common sensory impairments and heterogeneous disorders in humans. Up to 60% of HL cases are caused by genetic factors, and approximately 30% of genetic HL cases are syndromic. Although 400–700 genetic syndromes are associated with sensorineural HL (SNHL), caused due to problems in the nerve pathways from the cochlea to the brain, only about 45 genes are known to be associated with syndromic HL. Muscular dystrophies (MDs) are neuromuscular disorders characterized by progressive degeneration of skeletal muscle accompanied by non-muscular symptoms. MD-dystroglycanopathy (MD-DG), caused by aberrant glycosylation of α-dystroglycan, is an MD subtype with a wide spectrum of non-muscular symptoms. Despite a growing number of MD-DG subtypes (at least 18), no comprehensive study has investigated SNHL in MD-DG. Here, we found that hearing impairment was associated with abnormal myelination of the peripheral segment of the cochlear nerve caused by impaired dystrophin–dystroglycan complex in two mouse models (type 3 and 6) of MD-DG and in patients (type 4) with MD-DG. This is the first comprehensive study investigating SNHL in MD-DG. Our findings may provide new insights into understanding the pathogenic characteristics and mechanisms underlying inherited syndromic hearing impairment.

## Introduction

Hearing loss (HL) is one of the most common sensory impairments and one of the most etiologically and genetically heterogeneous disorders in humans [1]. Congenital HL affects around 1 in 1000 live births, and an additional 1 in 1000 children will suffer from HL before adulthood [2]. Up to 60% of HL cases are caused by genetic factors [1], and approximately 70% and 30% of genetic HL cases are non-syndromic and syndromic, respectively [1, 3, 4]. Based on the site of the lesion electrophysiological tests, HL is categorized into conductive, sensorineural, mixed, and central HL, as well as auditory neuropathy [1, 5]. In the past two decades, extensive studies on the genetics of hereditary HL have been conducted, implicating 121 causative genes in non-syndromic sensorineural HL (SNHL) at around 150 SNHL loci (http://hereditaryhearingloss.org/) [4]. Although 400–700 genetic syndromes are accompanied by SNHL [3, 4], only about 45 genes are known to be associated with syndromic HL [3]. Syndromic HL often exhibits inconsistencies in the severity of HL and the age of onset; these inconstancies exists both between and within families [1].

Muscular dystrophies (MDs) are a heterogeneous group of genetic and neuromuscular disorders characterized by progressive degeneration of skeletal muscles accompanied by various non-muscular symptoms. They are caused by mutations in a large variety of genes, which encode proteins of the contractile apparatus, structural proteins, and post-translational modification enzymes; currently, more than fifty causative genes have been reported [6–8]. MDs are categorized based on factors such as causative genes, inheritance patterns, and clinical presentations. Duchenne MD is the most common form of MD, resulting from loss of functional dystrophin, a cytoplasmic actin-binding protein (Fig 1A). Several forms of MDs result at least partially from defects in the dystrophin–dystroglycan complex [8, 9], which serves to link the intracellular actin cytoskeleton to the extracellular matrix (basal lamina). Dystroglycan is a single gene (*dystrophin-associated glycoprotein 1*: *DAG1*) product that is processed into two subunits: β-dystroglycan (β-DG), a transmembrane protein that interacts with dystrophin in the cytoplasm, and α-dystroglycan (α-DG), a highly glycosylated protein that interacts with both

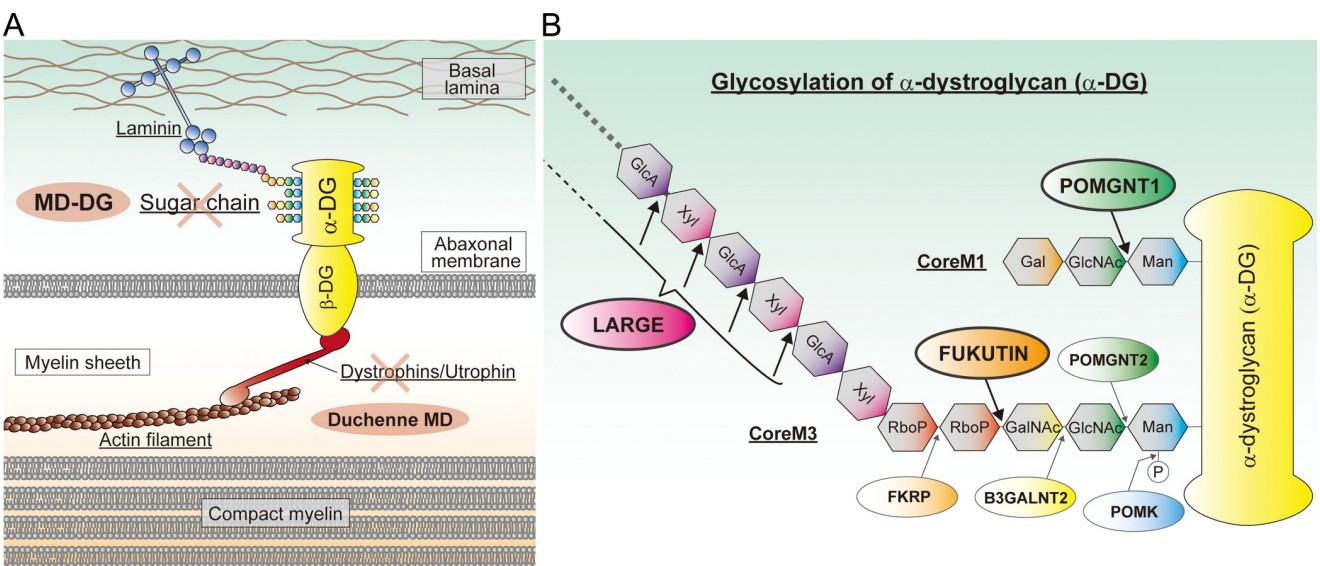

**Fig 1. The dystrophin–dystroglycan–laminin complex and the structure of the α-dystroglycan sugar chain. A,** Illustration showing the dystrophin–dystroglycan–laminin complex in peripheral nerve myelin. Mutations in enzymes modifying the α-dystroglycan (α-DG) sugar chain cause subtypes of muscular dystrophy known as muscular dystrophy-dystroglycanopathy (MD-DG), while dystrophin mutations cause Duchenne muscular dystrophy (Duchenne MD). **B,** Illustration showing the structure of the α-DG sugar chain and its modifying enzymes. α-DG is modified with *O*-mannose type glycans, namely CoreM1 and CoreM3. A repeating unit of GlcA-Xyl at the terminal of CoreM3 serves as a laminin-binding moiety. Enzymes referred to in the main text are shown, and their modification sites are indicated by arrows. Man: mannose, GlcNAc: *N*-acetylglucosamine, Gal: galactose, GalNAc: *N*-acetylgalacosamine, RboP: ribitol 5-phosphate, Xyl: xylose, GlcA: glucuronic acid.

β-DG and multiple components of the basal lamina, such as laminin (Fig 1A). Notably, proper glycosylation of α-DG is essential for its binding to the basal lamina.

A growing number of causative genes for congenital MD (CMD) with abnormal α-DG glycosylation, including *FUKUTIN, POMGNT1, POMGNT2, FUKUTIN RELATED PROTEIN* (*FKRP*), and *LARGE*, have been recognized, and to date, at least eighteen have been reported [8] (Fig 1B). Genetic abnormalities in *FUKUTIN* cause Fukuyama CMD, which is the second most common form of MD in Japan [10]. As aberrant α-DG glycosylation is a common biochemical feature, the diseases are called α-dystroglycanopathy or MD-dystroglycanopathy (MD-DG). However, MD-DG is clinically heterogeneous and exhibit a wide spectrum of symptoms, from combined brain and eye anomalies to only muscular phenotypes [11]. Most of the MD-DG causative genes encode enzymes that synthesize unique sugar chains of α-DG, namely CoreM1 and CoreM3 [8] (Fig 1B). LARGE is a bifunctional glycosyltransferase that synthesizes a repeating unit consisting of a glucuronic acid and a xylose, namely matriglycan, which provides the laminin-binding moiety on the terminal portion of CoreM3. FUKUTIN is an enzyme involving the synthesis of the tandem ribitol-phosphate structure, which serves as a fundamental unit required for LARGE-dependent matriglycan formation. POMGNT1 functions not only in the synthesis of CoreM1 but also plays important roles in regulating FUKUTIN function. Thus, a lack of either FUKUTIN or POMGNT1 consequently affects LARGE-dependent laminin-binding activity [9, 12].

SNHL has been reported in specific types of MD, such as facioscapulohumeral MD (FSHD) and myotonic dystrophy (DM: classified into DM1 and DM2). FSHDs (FSHD1 and FSHD2) are caused by aberrant expressions of DUX4 [13], and about 20% of FSHD1 patients show SNHL at high frequencies [13, 14]. A transgenic mouse model of FSHD has been generated [15], and the auditory brainstem response (ABR) tests show SNHL at frequencies greater than 8 kHz [16]; however, the underlying mechanism of SNHL has not been evaluated even in the

FSHD transgenic mice. DM1 is caused by an expanded CTG repeat in the 3' untranslated region of the gene encoding myotonic dystrophy protein kinase, and a high prevalence of SNHL (68%) has been reported in DM1 patients [17]. Furthermore, dysfunction of outer hair cells (OHCs) is implicated in SNHL in DM1 patients [18, 19], although how OHC function is impaired has not been clarified. Additionally, it remains unclear whether SNHL is present in $Dmd^{mdx}$ mice, a model of Duchenne MD (19, 20). Finally, although $Large^{myd}$ mice were proposed as a model of FSHD in the 1990s [20], $Large^{myd}$ mice were later proven to be a model of MD-DG type 6 [21]. Thus, molecular mechanisms underlying SNHL in FSHD and DM are ambiguous, and hearing function in MD-DG remains unclear despite the growing categories of MD-DG.

In the present study, we examined hearing function in $Large^{myd}$ and POMGnT1-KO mice, models of MD-DG, and patients with Fukuyama CMD, a type of MD-DG. We found hearing impairment in both MD-DG mouse models and in Fukuyama CMD patients. In the mouse models, profoundly abnormal myelination was observed at the peripheral segment of the cochlear nerve located at Rosenthal's canal (RC) and the osseous spiral lamina (OSL), where the cochlear nerve is myelinated by Schwann cells. Our findings indicate that MD-DG is associated with congenital, non-progressive retrocochlear hearing impairment.

## Results

### HL in *Large*-deficient and *POMGnT1*-KO mice

To evaluate hearing function in MD-DG, we firstly measured ABRs to click and tone-burst stimuli (8, 16, and 24 kHz) in 2-, 3-, 5-, 8-, and 10-week-old *Large*-deficient ($Large^{myd/myd}$) mice. ABR threshold was significantly higher in $Large^{myd/myd}$ mice regardless of the age and sound frequency, except for 24 kHz in those aged 5 weeks, than in control (WT) and heterozygous *Large*-deficient ($Large^{myd/wt}$) mice (Fig 2A). The hearing impairment in $Large^{myd/myd}$ mice was non-progressive. In $Large^{myd/myd}$ mice, the wave I latency, but not latencies between wave I and wave V (I-V) or III-V, was significantly delayed at the ages of 3 and 8 weeks, and the amplitude of wave I was significantly reduced at the age of 8 weeks compared with control mice (Fig 2B). Distortion product otoacoustic emission (DPOAE), which is used to estimate OHC function in the cochlea [22], was impaired in $Large^{myd/myd}$ mice aged 2–9 weeks compared with control mice (S1A Fig). Subsequently, we examined the morphology of the cochlea in $Large^{myd/myd}$ mice. Hematoxylin and eosin (HE) and phalloidin staining revealed no apparent morphological abnormality or HC loss in the cochleae in $Large^{myd/myd}$ mice at the age of 8 weeks (Fig 2C).

*POMGnT1*-KO mice, another MD-DG mouse model also showed non-progressive ABR impairment at all frequencies compared with control and heterozygous *POMGnT1*-KO mice, but the impairment was milder in *POMGnT1*-KO mice than in $Large^{myd/myd}$ mice (Fig 3A). In addition, the wave I latency was significantly delayed in *POMGnT1*-KO mice at the age of 4 weeks compared with control mice, but no difference was detected between *POMGnT1*-KO and control mice at the age of 10 weeks (Fig 3B). No abnormality was observed in DPOAE in *POMGnT1*-KO mice at the age of 6 weeks (S1B Fig). In addition, no apparent morphological abnormality or HC loss was observed in *POMGnT1*-KO mice (Fig 3C). These findings suggest that: 1) hearing function is impaired in both these mouse models of MD-DG; 2) the hearing impairment is already present and non-progressive at 2–4 weeks after birth, when hearing function has matured [23, 24]; and 3) the impairment degree is more in $Large^{myd/myd}$ mice (moderate) than in *POMGnT1*-KO mice (mild).

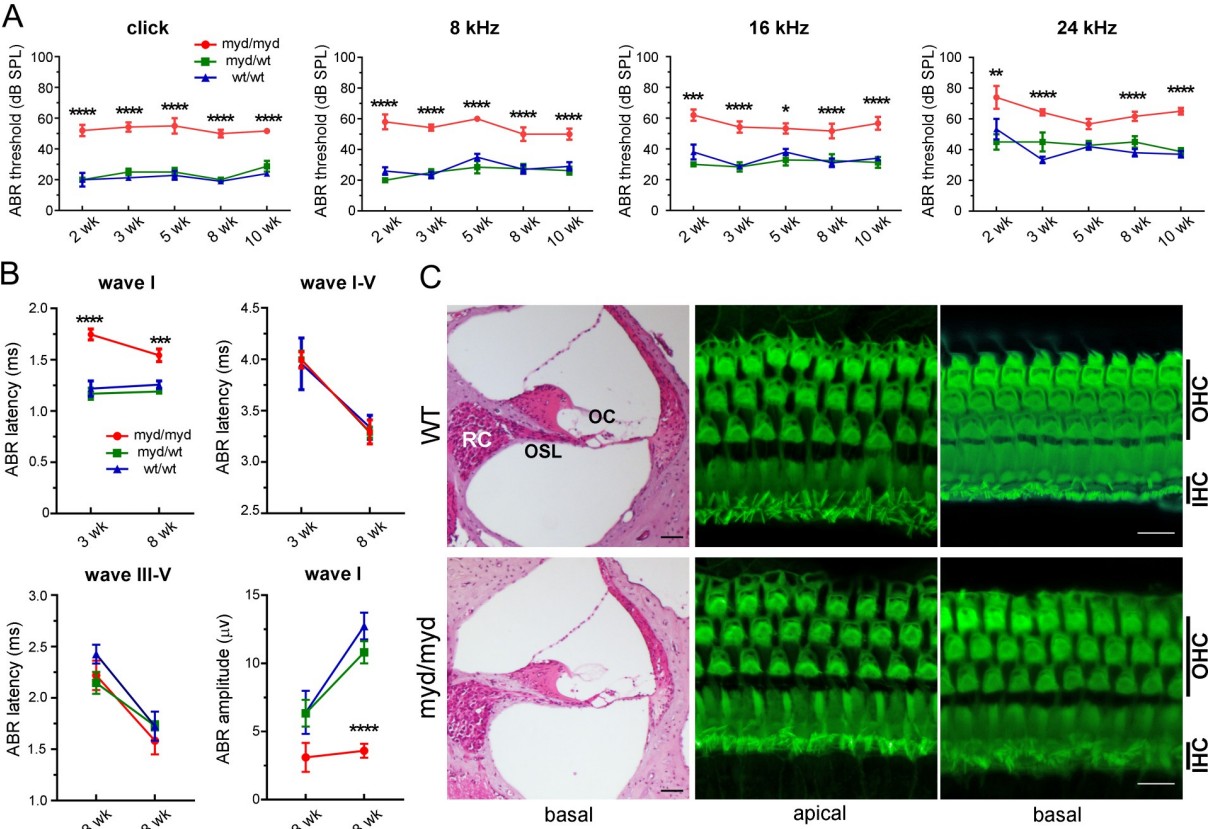

**Fig 2. Moderately impaired hearing in *Large^{myd/myd}* mice. A,** ABR thresholds (dB SPL) tested with clicks and pure-tone bursts at 8, 16, and 24 kHz in 2, 3, 5, 8, 10-week-old control (*Large^{wt/wt}*, $n$ = 3, 6, 5, 10, and 10, respectively), *Large^{myd/wt}* ($n$ = 3, 6, 7, 8, and 8, respectively), and *Large^{myd/myd}* mice ($n$ = 5, 7, 3, 6, and 6, respectively). Statistical significance was analyzed at all frequencies and ages between control and *Large^{myd/myd}* mice, except for 24 kHz frequency at 5 weeks of age. *$P$ < 0.05, **$P$ < 0.01, ***$P$ < 0.001, **** $P$ < 0.0001 using two-way ANOVA with Tukey's post-hoc test. **B,** ABR latencies of wave I, wave I-V, and wave III-V and the amplitude of wave I (click stimulation, 90 dB) in 3- and 8-week-old control ($n$ = 5 and 6–7, respectively), *Large^{myd/wt}* ($n$ = 5 and 5–6, respectively), and *Large^{myd/myd}* mice ($n$ = 6–7 and 6–7, respectively) were graphed. ****$P$ < 0.0001, ***$P$ = 0.0008 in the wave I latency and ****$P$ < 0.0001 in the wave I amplitude (control vs. *Large^{myd/myd}*) using two-way ANOVA with Tukey's post-hoc test. **C,** Inner ears of 8-week-old control and *Large^{myd/myd}* mice were fixed, and the cochleae were morphologically examined by HE staining at the basal turn and Alexa488-conjugated phalloidin staining at the apical and basal turns. No morphological abnormality in inner hair cells (IHCs), outer hair cells (OHCs), or hair cell loss was observed. The results were presented as the mean of three experiments. Scale bars: 50 μm for HE staining and 10 μm for phalloidin staining. RC: Rosenthal's canal; OSL: osseous spiral lamina; OC: organ of Corti.

## Decreased levels of α-DG glycosylation, laminin, and myelin basic protein (MBP) at the RC and OSL in *Large^{myd/myd}* and *POMGnT1*-KO mice

In the rodent inner ear, strong immunoreactivity of α-DG has been observed in the perineural basal lamina of the peripheral, but not central, segment of the cochlear nerve, which is clearly demarcated by "the glial dome" [25, 26]. The glial dome is located at the level of the basal turn of the cochlea and is the transitional zone of Schwann cells (peripheral) and oligodendrocytes (central) [27, 28]. Notably, α-DG is strongly positive at the RC and OSL, where cell bodies of spiral ganglion neurons (SGNs) are located and peripheral axons of SGNs (bipolar neurons) pass to reach hair cells (HCs). Moreover, greater degrees of glycosylated α-DG in the cochlea than in the cerebellum and brain have been demonstrated in rodents [25].

To detect the affected lesion in the cochlea in *Large^{myd/myd}* and *POMGnT1*-KO mice, we performed immunostaining using antibodies detecting the glycosylated form of α-DG, core α-DG

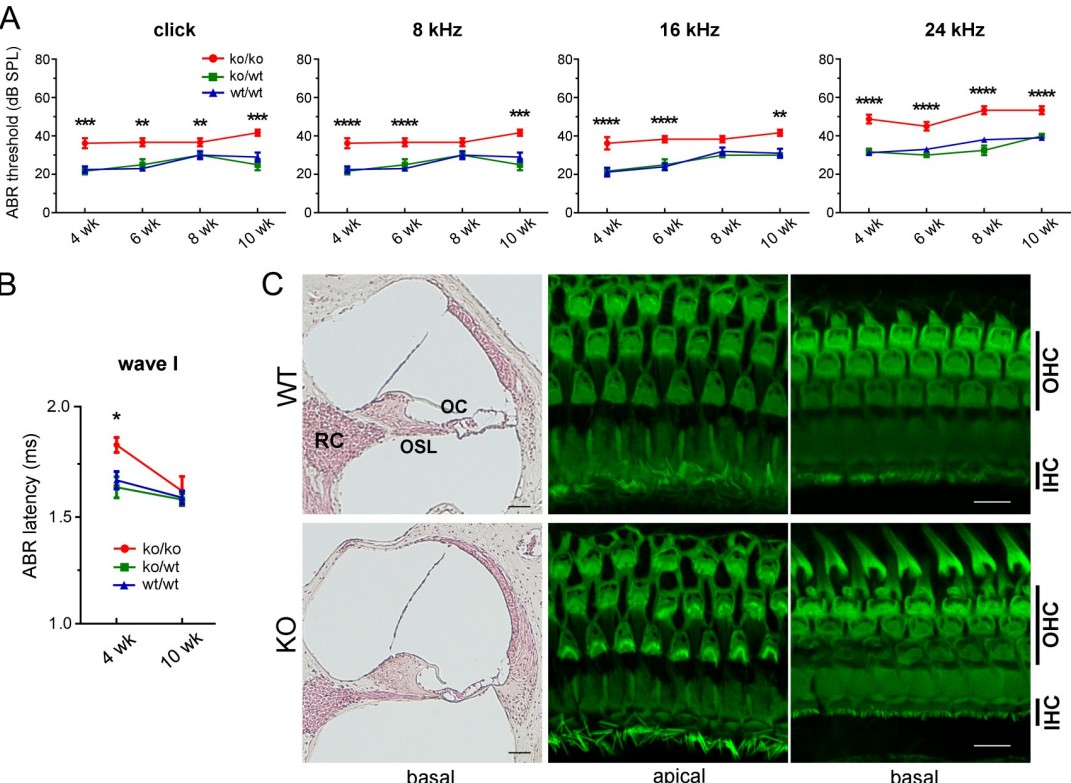

**Fig 3. Mildly impaired hearing in *POMGnT1*-KO mice. A,** ABR thresholds (dB SPL) tested with clicks and pure-tone bursts at 8, 16, and 24 kHz in 4, 6, 8, and 10-week-old control (*wt/wt*; *n* = 8, 10, 10, and 10, respectively), heterozygous *POMGnT1*-KO (*ko/wt*; *n* = 6, 4, 4, and 4, respectively), and *POMGnT1*-KO mice (*ko/ko*; *n* = 8, 6, 6, and 6, respectively). **P < 0.01, ***P < 0.001, **** P < 0.0001 (control vs. *POMGnT1*-KO) using two-way ANOVA with Tukey's post-hoc test. **B,** ABR latencies of wave I at the click stimulation at 90 dB in 4- and 10-week-old control (*n* = 6 and 7, respectively), heterozygous *POMGnT1*-KO (*n* = 7 and 4, respectively), and *POMGnT1*-KO (*n* = 7 and 4, respectively) mice were graphed. *P = 0.0197 (control vs KO) using two-way ANOVA with Tukey's post-hoc test. **C,** Inner ears of control and *POMGnT1*-KO mice were fixed and morphologically examined by HE staining (8-week-old) and Alexa488-conjugated phalloidin staining (12-week-old). No morphological abnormality in inner hair cells (IHCs), outer HC (OHCs), or hair cell loss was observed. The results were presented as the mean of three experiments. Scale bars: 50 μm for HE staining and 10 μm for phalloidin staining. RC: Rosenthal's canal; OSL: osseous spiral lamina; OC: organ of Corti.

protein, β-DG, pan-laminin, or laminin α2. Although immunoreactivities of core α-DG and β-DG did not differ between control and *Large*$^{myd/myd}$ mice, those of glycosylated α-DG, pan-laminin, and laminin α2 significantly decreased at the peripheral portion of the cochlear nerve (distal from the glial dome), especially at the RC and OSL, in *Large*$^{myd/myd}$ mice compared with control mice (Fig 4A–4C). Immunoreactivity of glycosylated α-DG, but not core α-DG or β-DG, was also decreased in *POMGnT1*-KO mice compared with control mice (Fig 5A and 5B). Furthermore, the immunoreactivity of laminin α2 was mildly but significantly decreased (Fig 5A and 5B). We were unable to show the decreased levels of glycosylated α-DG or laminin α2 by immunoblotting because of the limitation of the antibodies. However, we detected decreased levels of pan-laminin by immunoblotting using lysates obtained from the spiral ganglion (SG), which is located in the RC, in P5-P7 *Large*$^{myd/myd}$ mice (Fig 5C). In *POMGnT1*-KO mice, the pan-laminin levels also tended to be decreased compared with the controls, but in a limited sample number (Fig 5D, *n* = 1 in WT, *n* = 2 in heterozygous *POMGnT1*-KO, and *n* = 2 in *POMGnT1*-KO).

Subsequently, to investigate the underlying molecular mechanism of the lesions, we examined the myelination of the cochlear nerve at the RC and OSL using immunohistochemistry

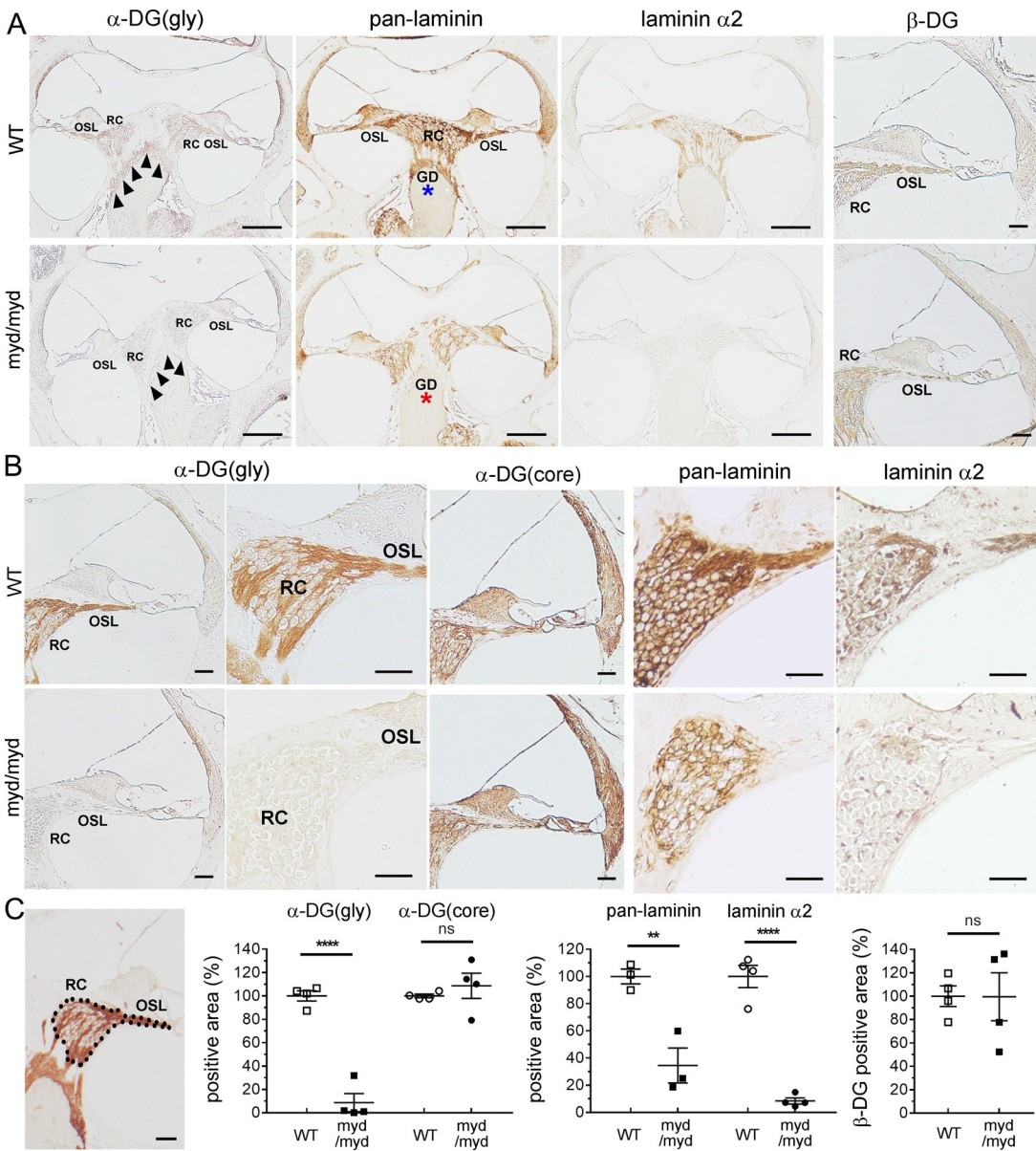

**Fig 4. Decreased glycosylated α-DG and laminin levels distal to the glial dome in *Large^{myd/myd}* mice.** Inner ears of 8-week-old control and *Large^{myd/myd}* mice (**A–C**) were fixed. **A,** Immunostaining of glycosylated α-dystroglycan [α-DG(gly)], pan-laminin, and laminin α2 at the basal turn level of the cochlea [glial dome (GD) is indicated by arrowheads]. Scale bars: 200 μm. Magnified immunostainings of β-dystroglycan (β-DG) at the basal turn of the cochlea are also shown (Scale bars: 50 μm). RC: Rosenthal's canal; OSL: osseous spiral lamina. Asterisks indicate the regions (proximal to the GD) used for TEM analysis (shown in S5 Fig). **B,** Magnified immunostainings of α-DG(gly) and core α-DG [α-DG(core)] at the basal turn of the cochlea. More magnified immunostained images of α-DG(gly), pan-laminin, and laminin α2 at the RC and OSL of the cochlear basal turn are shown. Scale bars: 50 μm. **C,** Illustration shows the area including the RC and OSL (surrounded by dots) used for quantitative analysis (Scale bars: 50 μm). Statistical analyses of α-DG(gly), α-DG(core), pan-laminin, laminin α2, and β-DG immunostainings at the RC and OSL. $n$ = 4, 4, 3, 4, and 4, respectively. ****$P$ < 0.0001, $P$ = 0.4587, **$P$ = 0.0093, ****$P$ < 0.0001, and $P$ = 0.9813, respectively (Student's $t$-test). ns: not significant.

with an antibody against MBP, a myelination marker. Immunoreactivity of MBP was decreased at the RC and OSL in *Large^{myd/myd}* mice (Fig 6A). The decreased levels of MBP were further confirmed by immunoblotting using lysates obtained from the SG of P5-P7 *Large^{myd/myd}* mice

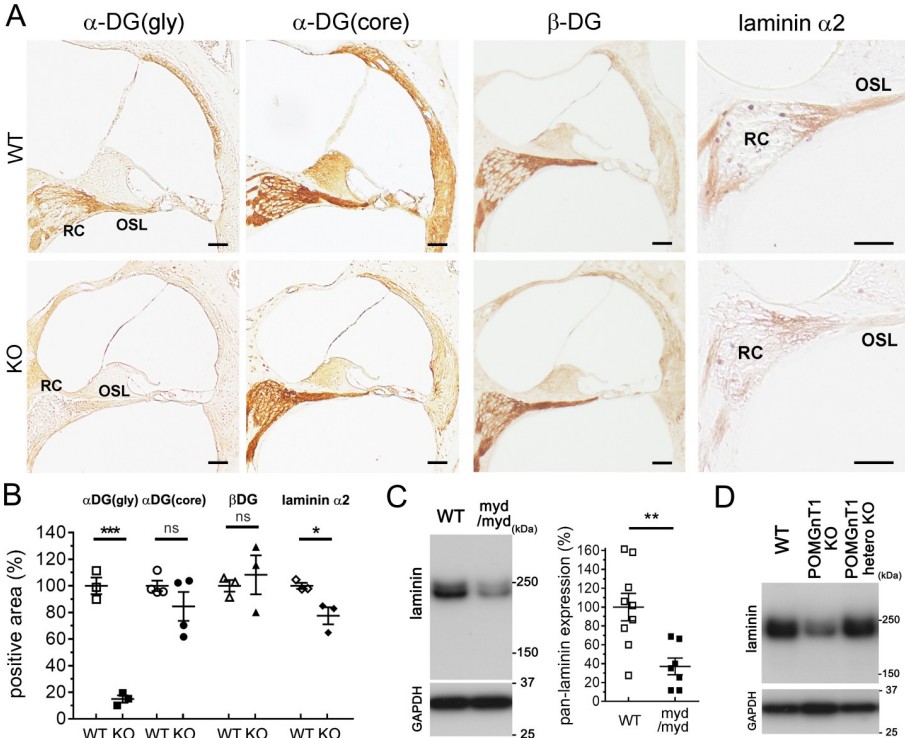

**Fig 5. Decreased levels of glycosylated α-DG and laminin levels distal to the glial dome in *POMGnT1*-KO and *Large^myd/myd* mice. A,** Inner ears of 16-week-old control and *POMGnT1*-KO mice were fixed. Glycosylated α-dystroglycan [α-DG(gly)], core α-DG [α-DG(core)], β-dystroglycan (β-DG), and laminin α2 immunostainings at the basal turn level of the cochlea. RC: Rosenthal's canal; OSL: osseous spiral lamina. Scale bars: 50 μm. **B,** Statistical analyses of α-DG(gly), α-DG(core), β-DG, and laminin α2 immunostainings at the RC and OSL. *n* = 3, 4, 3, and 3, respectively. ***P = 0.0002, P = 0.2274, P = 0.6166, and *P = 0.0294, respectively (Student's *t*-test). **C and D,** Pan-laminin levels in the spiral ganglion (SG) of P5-7 control and *Large^myd/myd* mice (**C**) and control, heterozygous *POMGnT1*-KO, and *POMGnT1*-KO mice (**D**) were detected by immunoblotting with GAPDH as loading controls. Statistical analysis was performed in pairs of control (*n* = 9) and *Large^myd/myd* mice (*n* = 7, P = 0.0040 using Student's *t*-test).

(Fig 6B). Decreased immunoreactivity and protein levels of MBP were also detected in *POMGnT1*-KO mice (Fig 6C and 6D), but the decreased levels were larger in *Large^myd/myd* mice than in *POMGnT1*-KO mice. The expression levels of 2, 3-cyclic nucleotide-3-phosphodiesterase (CNPase), another myelination marker, were also decreased in the *Large^myd/myd* mice (S2 Fig).

## No HL in *Dmd^mdx/mdx* mice

HL has been never reported in patients with Duchenne MD or Becker MD [29]. To confirm this finding, we examined HL in *Dmd^mdx/mdx* mice, a model of Duchenne MD. ABR and DPOAE of *Dmd^mdx/mdx* mice were comparable with those of control mice (S3A Fig). No differences were detected in immunoreactivities of core and glycosylated α-DG proteins (S3B Fig) or MBP between control and *Dmd^mdx/mdx* mice (S3C Fig). These findings strongly suggest that HL observed in two mouse models of MD-DG is a notable phenotype/symptom of MD-DG.

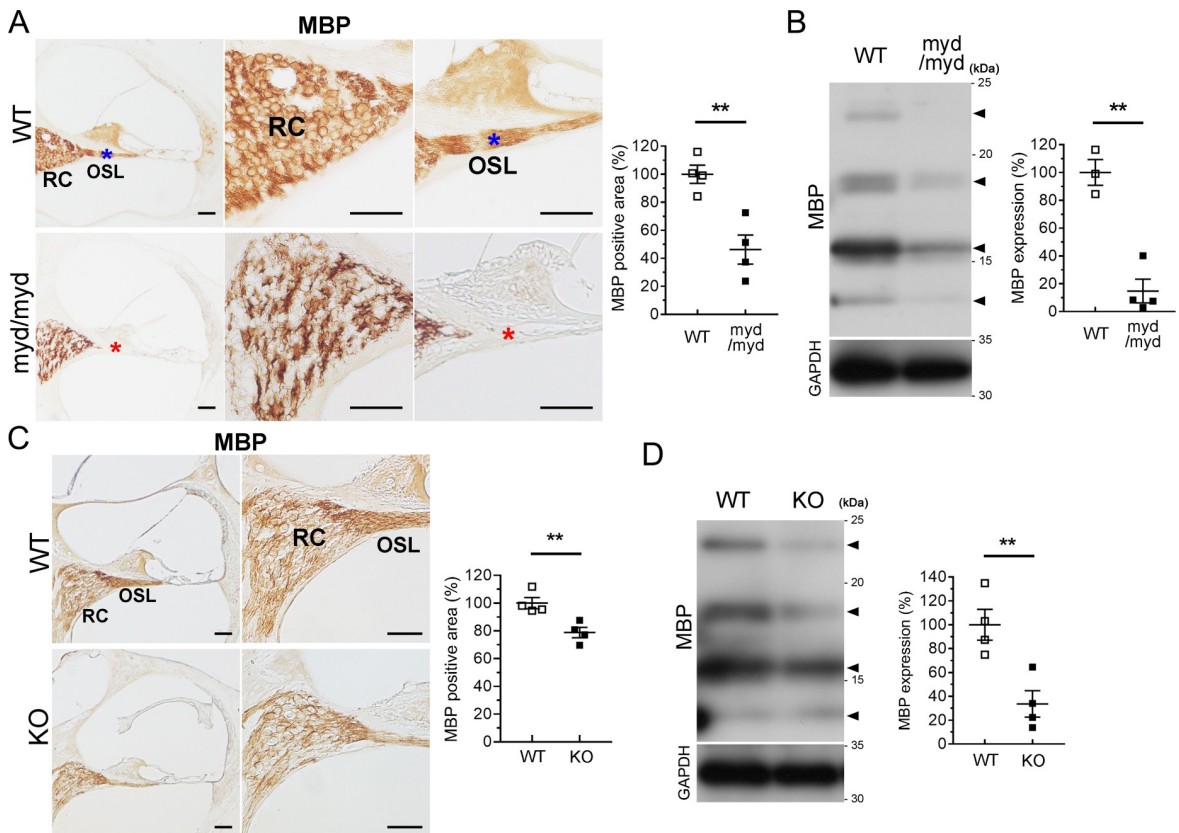

**Fig 6. Decreased MBP levels in *Large^{myd/myd}* and *POMGnT1*-KO mice at the RC and OSL.** Cochlear sections at the basal turn (9-week-old) and lysates from SG (P5-7) were obtained from control and *Large^{myd/myd}* mice (**A** and **B**). Cochlear sections at the basal turn (16-week-old) and lysates from SG (P5-7) were also obtained from control and *POMGnT1*-KO mice (**C** and **D**). Immunostaining (**A** and **C**) and immunoblotting (**B** and **D**) were performed using an MBP antibody, and statistical analyses were conducted using Student's *t*-test. Comparable loading of proteins was confirmed by immunoblotting of GAPDH. Scale bars: 50 μm. RC: Rosenthal's canal; OSL: osseous spiral lamina. Asterisks (A) indicate the regions (OSL) used for TEM analysis (shown in Figs 7 and S4). Arrowheads (**B** and **D**) indicate MBP variants used for quantitative analysis. **A,** *n* = 4 in each group, **P = 0.0046; **B,** *n* = 3 in control mice and 4 in *Large^{myd/myd}* mice, **P = 0.0011; **C,** *n* = 4 in each group, **P = 0.0086; **D,** *n* = 4 in each group, **P = 0.0080.

## Abnormal myelination at the peripheral segment of the cochlear nerve in both *Large^{myd/myd}* and *POMGnT1*-KO mice

In peripheral nerves, α-DG is expressed in the outer (abaxonal) membranes of the Schwann cells to bind the basal lamina [30–32]. Transmission electronic microscopy (TEM) analysis was used to examine myelination of the peripheral segment of the cochlear nerve, which is projected by Schwann cells [27], in *Large^{myd/myd}* mice at the age of 2, 6, and 10 weeks. *Large^{myd/myd}* mice showed various abnormalities, such as naked axons, axons with disrupted myelin, and axons with secondary changes (vacuoles and/or aggregates) (Figs 7A and S4A and S4B). The percentage of axons with abnormal myelination (naked axons and axons with disrupted myelin) at the OSL was decreased in *Large^{myd/myd}* mice aged 6 and 10 weeks compared with that in mice aged 2 weeks and those aged 2 and 6 weeks, respectively; however, this number was significantly higher in *Large^{myd/myd}* mice than in age-matched control mice aged 2–10 weeks (Fig 7B). The number of axons with secondary changes was significantly higher in *Large^{myd/myd}* mice than in controls 6 and 10, but not 2 weeks old (S4B Fig). In addition, diameters of myelinated axons in the transverse section at the OSL were more broadly distributed in *Large^{myd/myd}* mice than in control mice aged 6 weeks (Fig 7C), but it was not significantly different

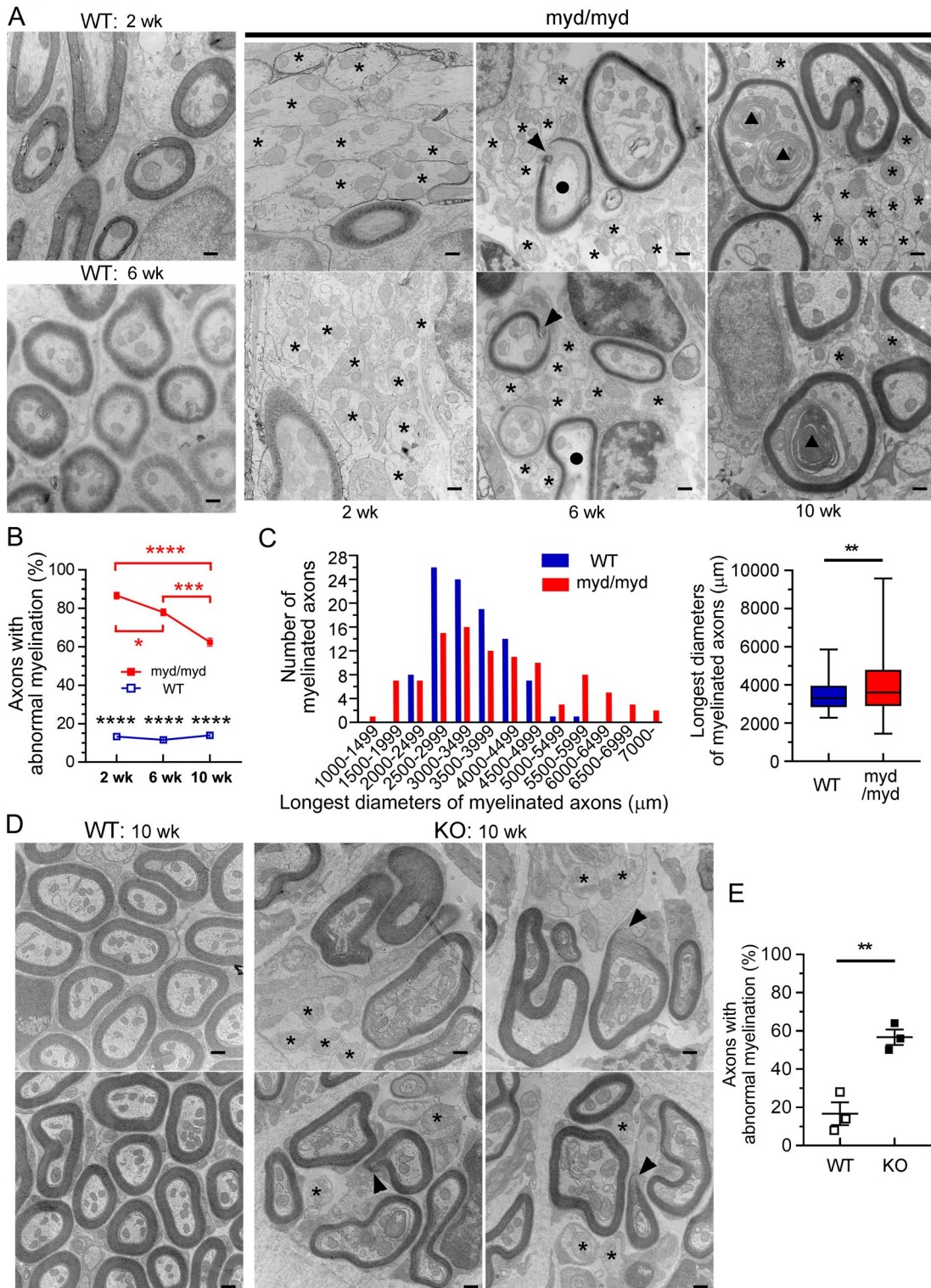

**Fig 7. Abnormal myelination of the peripheral cochlear nerve at the OSL in *Large^{myd/myd}* and *POMGnT1*-KO mice.** Inner ears of 2-, 6-, and 10-week-old control and *Large^{myd/myd}* mice (**A-C**) and 10-week-old control and *POMGnT1*-KO mice (**D**) were fixed for transmission electron microscopy (TEM). TEM images at the osseous spiral lamina (OSL, indicated by the asterisks in Fig 6A) were obtained. The asterisks, arrowheads, circles, and triangles indicate naked axons, axons with disrupted myelin, and myelinated axons with vacuoles or aggregates, respectively. Scale bars: 500 nm. The percentage (each obtained by analyzing 50

axons) of axons with abnormal myelination was statistically analyzed in 2-, 6-, and 10-week-old control and $Large^{myd/myd}$ mice (**B**) and in 10-week-old control and $POMGnT1$-KO mice (**E**). (**B**) $****P < 0.0001$ using two-way ANOVA with Bonferroni's post-hoc test (control vs. $Large^{myd/myd}$). $*P = 0.0399$ ($Large^{myd/myd}$ mice at 2 weeks vs. 6 weeks), $***P = 0.0003$ ($Large^{myd/myd}$ mice at 6 weeks vs. 10 weeks), and $P < 0.0001$ ($Large^{myd/myd}$ mice at 2 weeks vs. 10 weeks) using one-way ANOVA with Tukey's post-hoc test. $n = 3, 5$, and 5 in 2-, 6-, and 10-week-old control mice, respectively. $n = 3, 6$, and 5 in 2-, 6-, and 10-week-old $Large^{myd/myd}$ mice, respectively. (**E**) $n = 3$ in both groups, $**P = 0.0051$ using Student's $t$-test. **C,** Distribution of the longest diameter of each myelinated axon in the transverse section at the OSL in 6-week-old control and $Large^{myd/myd}$ mice are graphed and statistically analyzed. $n = 100$ from 3 cochleae (30, 30, and 40 in control mice and 29, 33, and 38 in $Large^{myd/myd}$ mice. $**P = 0.0063$ using Kolmogorov–Smirnov test.

between $Large^{myd/myd}$ mice aged 6 weeks and those aged 10 weeks (S5A Fig). No apparent difference was observed in the percentage of myelinated axons of the cochlear nerve proximal to the glial dome in $Large^{myd/myd}$ mice aged 5 weeks (S5B Fig). However, the percentage of axons with abnormal myelination was mildly higher in $Large^{myd/myd}$ mice than in control mice (S5B Fig), suggesting that lesions proximal to the glial dome is milder than those distal to the glial dome. No significant difference in total myelination proximal to the glial dome between $Large^{myd/myd}$ and control mice was confirmed by MBP immunostaining (S5C Fig). Although the difference was smaller in $POMGnT1$-KO mice than in $Large^{myd/myd}$ mice, the percentage of axons with abnormal myelination at the OSL was significantly higher in $POMGnT1$-KO mice than in control mice at the age of 10 weeks (Figs 7D and 7E and S4C and S4D). These findings suggest that the main lesion in MD-DG model mice is localized at the peripheral cochlear nerves distal to the glial dome.

Furthermore, MBP immunoreactivity at the corpus callosum was examined to determine the myelination status in the brain. MBP immunoreactivity was mildly decreased in $Large^{myd/myd}$ mice compared with control mice at the age of 8 weeks (S6A Fig). This finding was confirmed by MBP immunoblotting using whole-brain lysates from P7 mice (S6B Fig).

## Fukuyama CMD patients showed delayed latency of wave I in ABR

Based on the findings obtained from the two mouse models of MD-DG, we hypothesized that MD-DG patients have the primary lesion at the peripheral segment of the cochlear nerve located at the RC and OSL, thus leading to retrocochlear SNHL. To test our hypothesis, we attempted to analyze auditory function in patients with Fukuyama CMD, which is the most frequent MD-DG in Japan. None of the Fukuyama CMD patients analyzed had complaint of deafness (Table 1). Because of mild to moderate intellectual impairment, a subjective audiometric examination was not applicable, and we performed the ABR analysis instead.

ABR response to 40-dB SPL stimuli were detected in 17 ears among 18 ears in 9 Fukuyama CMD patients, which was recognized by the presence of wave V. Only a 15-month-old boy with homozygous SINE-VNTR-$Alu$ (SVA) mutation (Case 1 in S1 Table) showed no recognizable ABR waveform in his left ear with 60-dB SPL stimuli. However, we could not discriminate whether the undetectable ABR was caused by conductive HL or SNHL; thus, this ear was excluded from the following analysis. In addition, ABRs to 40-dB SPL stimuli were detected in all 18 ears in 9 normal healthy volunteers (S2 Table).

Next, we analyzed the amplitude and latency of ABR waves above the thresholds to examine the possibility of latent hearing impairment. The latency of wave I at 60 dB ($2.03 \pm 0.32$ ms) was significantly delayed in the ears in Fukuyama CMD patients with recognizable ABR compared with normal controls ($1.62 \pm 0.15$ ms, $P < 0.0001$) (Fig 8A and S3 Table). This significant difference in the wave I latency was confirmed in two more detailed analyses, in which Fukuyama CMD patients were divided into the following two groups according to the genetic abnormality: one with SVA homozygous mutation and the other with compound

**Table 1. Clinical characteristics of Fukuyama CMD patients evaluated in the present study.**

| case | genotype | months | sex | severity | involved organs |
|---|---|---|---|---|---|
| 1 | SVA Homozygous | 15 | M | mild | CC and mild ID |
| 2 | SVA Homozygous | 94 | F | moderate | CC and mild ID |
| 3 | SVA Homozygous | 198 | M | moderate | CC, moderate ID epilepsy, scoliosis, and cardiomyopathy |
| 4 | SVA Homozygous | 239 | F | mild | CC and mild ID |
| 5 | SVA/Heterozygous Ex3 | 54 | F | moderate | CC, moderate ID, myopia, and dysphagia |
| 6 | SVA/Heterozygous Ex3 | 17 | M | moderate | CC and moderate ID |
| 7 | SVA/Heterozygous Ex3 | 66 | F | moderate | CC, myopia, moderate ID, and myopia |
| 8 | SVA/Heterozygous Ex3 | 74 | F | moderate | CC, myopia, moderate ID, and myopia |
| 9 | SVA/Heterozygous Int5 | 151 | M | severe | lissencephaly, CC, retinal detachment, myopia, cataracts, severe ID, cardiac arrest, and tracheotomy |

A total of nine Fukuyama CMD patients were analyzed in the study; genotype, months after birth, sex, severity of muscular dystrophy, and involved organs are indicated. Severity was classified based on the physical activity: mild, able to crawl; moderate, able to sit; severe, unable to control head position. CC, cerebellar cyst; ID, intellectual disability.

heterozygous mutation. Significantly delayed latency of wave I was detected in both homozygous patients ($2.07 \pm 0.24$ ms, $P = 0.0006$) and compound heterozygous patients ($2.00 \pm 0.37$ ms, $P = 0.0010$) compared with normal volunteers (homozygous, $1.57 \pm 0.15$ ms and heterozygous, $1.66 \pm 0.15$ ms) (Fig 8B and 8C and S4 and S5 Tables). The amplitude of wave I and the interpeak latency between waves I and V were not significantly different in all three analyses (Fig 8A–8C and S3–S5 Tables). These clinical findings suggest that Fukuyama CMD patients have latent/subclinical SNHL.

## Discussion

The present study indicates that *Large*$^{myd/myd}$ and *POMGnT1*-KO mice, models of MD-DG, exhibit congenital hearing impairment, and the lesions are localized at the peripheral segment of the cochlear nerve, where myelination is projected by Schwann cells, but not oligodendrocytes [27]. Large and POMGnT1 are expressed ubiquitously, including in skeletal muscles; however, very few reports have investigated their spatio-temporal expression patterns. *Large* is also highly expressed in the nervous system from the embryonic stage, including in the cerebral cortex and trigeminal ganglion [33, 34]; in contrast, expression of *POMGnT1* in the brain is relatively low [35]. Although POMGnT1 and Fukutin expression has been reported in astrocytes, Large, POMGnT1, or Fukutin expression in Schwann cells has not been reported [36]. In peripheral nerves, α-DG is expressed in the outer (abaxonal) membranes of Schwann cells to bind to basal lamina proteins, such as laminin α2 [30–32]. The mechanism of myelin formation, in which the anchorage of the abaxonal membranes to the basal lamina provided by the α-DG–laminin complex enables the inner lips of spiraling Schwann cells to progress over the axonal membranes, is well studied [30, 37, 38]. Impairment of peripheral nerve myelination causing various axonal abnormalities, including naked axons and axons with abnormal myelin, was reported in *Large*$^{myd/myd}$ mice [39], *Fukutin*-deficient chimeric mice [40], a patient with *LAMA2* deficiency [41], and a Fukuyama CMD patient [42]. Moreover, the dystroglycan–laminin complex is reportedly more important for myelination/differentiation of Schwann cells than for their survival/proliferation [43]. The findings in these studies support our results that α-DG and laminin α2 are critical for myelin formation in the peripheral segment of the cochlear nerve.

The mechanisms underlying abnormal DPOAE in *Large*$^{myd/myd}$ mice, but not in *POMGnT1*-KO mice, are currently unknown because we were unable to detect morphological

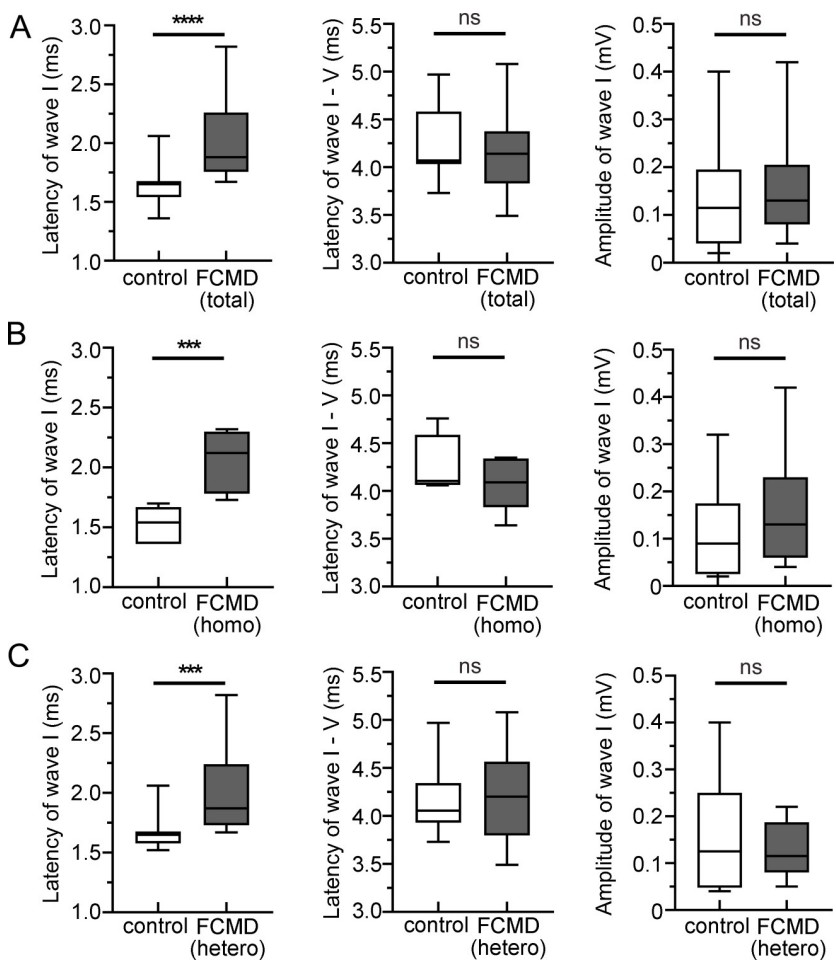

**Fig 8. Delayed latency of wave I of ABR in Fukuyama CMD patients. A**, Hearing function of Fukuyama CMD patients (total, $n = 17$; homozygous SVA-insertion, $n = 7$; compound heterozygous mutation with a SVA-insertion, $n = 10$) was analyzed using ABR. Significantly delayed latency of wave I was observed in analysis using all Fukuyama CMD patients (**A**), homozygous (homo) patients (**B**), and heterozygous (hetero) patients (**C**) compared with age- and sex-matched controls ($n = 18$, 8, and 10, respectively; ****$P < 0.0001$, ***$P = 0.0006$, and ***$P = 0.0010$, respectively, using Mann-Whitney's U-test). Interpeak latency between wave I and V (**A–C**) and wave I amplitude (**A–C**) was not different between controls and Fukuyama CMD patients ($P = 0.8005$ and $P = 0.3902$ in all Fukuyama CMD patients, $P = 0.3470$ and $P = 0.2945$ in homo patients, and $P = 0.7542$ and 0.8677 in hetero patients, respectively, using Mann-Whitney's U-test).

abnormality of OHCs or connective spaces between OHCs and supporting cells or between OHCs and underlying basal lamina, where α-DG is reportedly expressed [25, 26], in $Large^{myd/myd}$ mice (S7 Fig). Since MD-DG is a complex disease, which causes lesions in many organs and tissues, including the peripheral nerves, brainstem, and brain, one simple explanation is that the efferent fibers in the cochlear nerve, relaying the feedback from cortico-olivocochlear pathways, which is proposed to modulate otoacoustic emission (OAE) generation [44], could be more severely affected in $Large^{myd/myd}$ than in $POMGnT1$-KO mice. Although abnormal cortico-olivocochlear efferent function is associated with various auditory disorders, such as hyperacusis, tinnitus, and poor speech-in-noise recognition [45, 46], its effects on OAE are controversial; its activation decreases and enhances OAE levels depending on stimuli and conditions [46, 47]. Another possibility is that cochleae in $Large^{myd/myd}$ mice may have functional issues, but not morphological ones. Since α-DG is reported to be widely expressed in rodent

and human cochlea [25, 26, 48], including in the stria vascularis where endocochlear potential is generated, MD-DG dysfunction may affect cochlear function due to reduced endocochlear potential. Indeed, mild morphological atrophy was reported in the stria vascularis in Charcot-Marie-Tooth (CMT) disease type 1B (CMT1B), which is caused by congenital anomalies in myelination by Schwann cells [49, 50]. Additionally, because SNHL in DM1 shows abnormal OAE [18, 19], the same mechanism may underlie the dysfunction of $Large^{myd/myd}$ cochlea.

We found that $POMGnT1$-KO mice showed milder phenotypes than did $Large^{myd/myd}$ mice. There are three possible explanations: 1) $POMGnT1$-KO mice show the detectable amount of properly glycosylated α-DG [51], whereas properly glycosylated α-DG is completely absent in $Large^{myd/myd}$ mice [52]. This difference might be due to different enzyme functions: Large is an enzyme responsible for synthesizing laminin-binding matriglycan, whereas POMGnT1 acts as a regulatory enzyme for matriglycan formation [8]; 2) more than 60% of $POMGnT1$-KO mice usually die within 3 weeks after birth [53]; notably, the present study used mice aged ≥ 4 weeks, which probably had milder phenotypes [53, 54]. In fact, there is broad phenotypic variability in patients with abnormal $POMGNT1$ gene [55–57]; and 3) the genetic backgrounds of $Large^{myd/myd}$ and $POMGnT1$-KO mice are different (C57B/6 and 129SvEv, respectively). Regarding the recovery of wave I latency observed in 10-week-old $POMGnT1$-KO mice, they showed fewer secondary axonal changes than $Large^{myd/myd}$ mice (see S4B and S4D Fig), suggesting that, judged on the basis of morphological phenotypes, surviving $POMGnT1$-KO mice may have a greater potential for functional recovery than $Large^{myd/myd}$ mice. Alternatively, in addition to lesions at the peripheral segment of the cochlear nerve, $POMGnT1$-KO mice may have other lesions, which affect the early hearing phenotype but improve by 10 weeks of age.

Whether MD-DG pathology can lead to SNHL in humans remains an open question. Recently, mild to moderate HL was found in two patients (siblings) with homozygous truncating mutation in $O$-mannosyl kinase (POMK, Fig 1B), which is required for α-DG glycosylation, and this disease was classified as MD-DG 12 [58]. Among 9 reported patients in 6 families with mutations in $POMK$ [59–62], SNHL was demonstrated in the above mentioned two patients. Among 23 reported patients [63–65] with mutations in $B3GALNT2$ (MD-DG 11, Fig 1B), one patient [66] had SNHL. Six patients in four families with $LARGE$ mutations were reported, but hearing function based on intensive evaluation was not described [21, 67–69]. In patients with $POMGNT1$ mutations, symptoms associated with SNHL have not been investigated to date [55–57], and no Fukuyama CMD patient with SNHL has been reported either. The Fukuyama CMD patients showed prolonged ABR wave I latency, but their hearing thresholds are less than 40 dB, which is the minimum sound intensity used in this study. Nevertheless, prolonged wave I latency may cause minor hearing symptoms such as tinnitus, since several studies suggested that prolonged latency and reduced amplitude of wave I are characteristic findings for tinnitus patients with normal hearing [70]. Recently, "hidden HL" has been referred to as hearing dysfunction that cannot detected by standard tests of auditory thresholds, but can be diagnosed by a reduced wave I amplitude in the absence of ABR threshold or latency changes [71]. In addition to cochlear synaptopathy, which is induced by impaired synaptic connection between HCs and the cochlear nerve [72], peripheral neuropathy including demyelination of the cochlear nerve has been reported as a cause of "hidden HL" [71]. Demyelination-related hidden HL reportedly had an additional phenotype of permanently increased wave I latency in Guillain-Barre syndrome [73]. Additionally, demyelination-related hidden HL with abnormal speech recognition ability in noisy backgrounds, but normal ABR amplitude and latency, was reported in CMT type 1A (CMT1A) [74].

CMT1A and CMT1B are neuropathies caused by genetic abnormalities in peripheral myelin protein 22 and myelin protein zero, respectively, which are the main myelination-

associated proteins in Schwann cells, and commonly show progressive SNHL starting in adolescence [75]. Although Kovach et al. and Starr et al. reported abnormal ABR accompanied by decreased or absent OAE [49, 76], the principal pathology of CMT1B is myelin damage of the cochlear nerve initiating distally (accompanied by secondary damage of the cochlear nerve axons) without IHC loss [49, 50]. While children with CMT1A showed no prolonged latency of wave I [75], adult patients showed prolonged latency of wave I [77]. Thus, both CMT1 and MD-DG cause primary lesions at the distal portion of the cochlear nerve; however, CMT1 shows a progressive phenotype [75]. This difference is most likely to arise due to from the different etiologies of CMT1 and MD-DG (autosomal dominant vs. autosomal recessive inheritance, respectively), and the degree of cochlear synaptopathy involvement may also be different in the two diseases. Possible dysmyelination at the peripheral cochlear nerve in Fukuyama CMD patients may accompany cochlear synaptopathy, and may cause similar symptoms of "hidden HL", including tinnitus and hyperacusis. These pathologies that are otherwise missed in routine clinical examination could be detected by more intensive hearing assays, such as speech recognition test in noisy backgrounds and estimation of auditory temporal resolution/processing [74, 75, 78], suggesting that other types of MD-DGs also cause previously unrecognized common and specific retrocochlear pathologies with widely varying degrees of hearing impairment.

Fukuyama CMD is characterized by the combination of MD and dysgenesis of the central nerve system (CNS). However, a good mouse model of Fukuyama CMD is not available. Mice with knock-in of the SVA-transposal insertion, the most frequent mutation in Japanese Fukuyama CMD patients, show no phenotype because of the only 50% binding reduction in laminin [51], and conventional *Fukutin*-KO mice are embryonic lethal [79]. Brain MRI of Fukuyama CMD patients demonstrates that cerebral cortical dysplasia is the most frequently detected anomalies (97/207 patients), and white matter abnormalities are the second detected anomalies (35/207 patients) [10]. To date, it is unclear whether white matter lesions are involved in delayed myelination or dysmyelination. Delayed myelination is supported by the observation that white matter lesions lessen with age (usually until 5-year-old) [80]. In addition, other studies support dysmyelination [81] or/and hypomyelination [82] based on the following observations: 1) inconsistent distribution of white matter lesions with the time-course of myelination process, 2) no disappearing of white matter lesions in some patients (elder than 5-year-old) [80, 83, 84], and 3) data using MR spectroscopy [85]. Taken together, white matter lesions are not simply delayed myelination, but dysmyelination or/and hypomyelination is more or less involved. Indeed, we detected mildly decreased levels of MBP in the brain and the proximal cochlear nerve of *Large*$^{myd/myd}$ mice. In addition to the white matter lesions, Fukuyama CMD patients have dysplasia in the brain, which may also affect hearing function. However, we detected the prolonged latency of wave I with normal interpeak latency between waves I and V in ABR in *Large*$^{myd/myd}$ mice, as well as in Fukuyama CMD patients, indicating that the main lesions for SNHL in MD-DG are at the peripheral segment of the cochlear nerve myelinated by Schwann cells, rather than in the brain.

No HL in Duchenne/Becker MD patients [29, 86], as well as *Dmd*$^{mdx/mdx}$ mice [87], may be due to compensations by other types of dystrophins and/or utrophin, a homolog of dystrophin [88]. Indeed, specific dystrophin (Dp116) and utrophin are the major subtypes of dystrophins in Schwann cells [30, 88–90]; furthermore, Duchenne MD patients with Dp116 deficiency have no apparent peripheral neuropathy [86, 91]. In addition, agrin, as well as laminin 2, has been shown as the binding partners of α-DG in Schwann cells [92]. These diversities of dystrophin–dystroglycan complex in Schwann cells may account for the milder hearing phenotypes in Fukuyama CMD patients than in two mouse models.

The present study indicates that both animal models and patients of MD-DG have impaired myelination at the peripheral segment of the cochlear nerve (at the RC and OSL), which potentially causes retrocochlear/retrolabyrinthine hearing impairment. Other types of MDs, including Duchenne/Becker MD, may also have the cryptic lesions at the peripheral segment of the cochlear nerve, leading to various types of hearing impairments/symptoms such as HL, "hidden HL", hyperacusis, and tinnitus. However, we did not find correlations between genotypes (homozygous vs. heterozygous mutation, which generally predict severity of MD phenotypes) and hearing phenotype (prolonged latency of wave I in ABR) in Fukuyama CMD patients. Studies with larger sample sizes are needed to further investigate HL in diseases with impairments of the dystrophin–dystroglycan complex.

## Methods

### Ethics statement

All procedures and experiments were reviewed and approved by the Institutional Review Board of Kobe University (clinical research number, 1653; animal research number, 26-03-05 and P150201-R3) and were performed in accordance with the ethical standards stated in the 1964 Declaration of Helsinki. Written informed consent was received from participants prior to inclusion in the study.

### Human studies

Nine Fukuyama CMD patients (four males and five females) aged 1–19 years and age-matched normal volunteers participated in the study. The diagnosis of Fukuyama CMD patients was confirmed by genetic analysis in all patients from peripheral blood. All patients with Fukuyama CMD had an ancestral SVA mutation in one allele [93]. Four patients (Case 1–4, Table 1) had homozygous SVA mutations. The compound heterozygous mutations in patients with Fukuyama CMD included a nonsense mutation in exon 3 [94] in four patients (Cases 5–8, Table 1) and a deep intronic splicing mutation in intron 5 [95] in one patient (Case 9, Table 1). Generally, phenotypes of Fukuyama CMD are more severe in compound heterozygous patients than in those with homozygous SVA mutations [10, 94].

### Animals

*Large*-deficient (*Large^{myd/myd}*) mice were obtained from Jackson Laboratories. *POMGnT1*-KO mice were bred in house as previously described [53]. *Dystrophin*-deficient mdx (*Dmd^{mdx}*) mice were obtained from CLEA Japan. Offspring were genotyped using PCR with the following primer pairs: *Large^{myd}*: 5′–GGCCGTGTTCCATAAGTTCAA–3′ and 5′–GGCATACGCC TCTGTGAAAAC–3′ for wild type (WT) and 5′–ATCTCAGCTCCAAAGGGTGAAG–3′ and 5′–GCCAATGTAAAATGAGGGGAAA–3′ for mutant; *POMGnT1*: 5′–CAGCAGTTTCCTT CCTTCTAACCC–3′ and 5′–ATTTGGTCTGGTCCCTTGGCTC–3′ for WT and 5′–AGGCT ATTCGGCTATGACTGGG–3′ and 5′–TACTTTCTCGGCAGGAGCAAGGTG–3′ for mutant; *Dmd^{mdx}*: 5′–TCATCAAATATGCGTGTTAGTG–3′ for both WT and mutant, 5′–GTCACTCA GATAGTTGAAGCCATTTAG–3′ for WT and 5′–GTCACTCAGATAGTTGAAGCCATTTAA–3′ for mutant. All mice were identified by numbered ear tags. Mice were housed in specific pathogen-free conditions using the individually ventilated cage system (Techniplast, Tokyo, Japan), with food and water ad libitum. The animal facility was maintained on a 14-h light and 10-h dark cycle at $23 \pm 2°C$ and $50 \pm 10\%$ humidity. Both males and females were used in the analyses unless otherwise indicated. Age-and sex-matched WT siblings were used as controls (mice younger than 1 week were not differentiated based on sex).

## Antibodies and chemicals

The following specific antibodies were used for both immunoblotting (IB) and/or immunohistochemistry (IH) (polyclonal unless indicated): monoclonal MBP (ab40390, Abcam, RRID; AB_1141521, 1/750 for IB and 1/250 for IH), CNPase (D83E10) XP (Cell Signaling Technology (CST), RRID; AB_10705455, 1/750 for IB), core α-DG (3D7; 1/100 for IH, [52]), glycosylated α-DG (IIH6C4, Millipore, RRID; AB_309828, 1/100 for IH), β-DG (8D5, Novocastra; 1/100 for IH), laminin (L9393, Sigma-Aldrich, RRID; AB_477163, 1/1000 for IB and 1/100 for IH), and laminin α2 (4H8-2, Enzo, RRID; AB_2051764, 1/50 for IH). HRP-conjugated antibodies against GAPDH (M171-7, 1:20000) and Alexa Fluor 488-conjugated phalloidin (1/1000) were used as described previously [96]. HE solutions were obtained from Muto Pure Chemicals (Japan).

## Histocytochemistry

Dissected tissues were fixed with 4% paraformaldehyde in 0.1 M phosphate buffer (PB, pH 7.4) and decalcified in 0.12 M EDTA for 1 week at 4˚C to examine surface preparations of the cochleae [24]. After permeabilization, fixed tissues were incubated with Alexa Fluor 488-conjugated phalloidin for 2 h at 23˚C. Stained tissues were mounted with Prolong anti-fade (Invitrogen) and were observed under an LSM700 confocal microscope (Carl Zeiss, Jena, Germany).

Tissues were isolated from indicated mice and fixed with 4% paraformaldehyde in 0.1 M PB to perform cross-section analyses of cochleae. Decalcified cochleae were embedded into paraffin blocks and cut into 6-μm slices on a Leica RM2125 RTS manual rotary microtome (Leica Biosystems, Wetzlar, Germany). Sections were stained (HE staining or immunostaining) after deparaffinization. To analyze cryosections of the brains, adult mice were transcardially perfused with ice-cold 0.9% saline solution and subsequently with 4% paraformaldehyde in 0.1 M PB. For antigen unmasking, the slides were bathed in HistoVT One (Nacalai Tesque, Kyoto, Japan) for 20 min at 80˚C. Retrieved tissues were blocked in either 5% fat-free BSA and 3% $H_2O_2$ (Nacalai Tesque) or 0.1% phenylhydrazine (Nacalai Tesque) for 20 min at 23˚C. The tissues were incubated with primary antibodies for 2 h at 23˚C, followed by MACH 2 Universal HRP-Polymer Detection (BIOCARE Medical, Pacheco, CA, USA) for 30 min at 23˚C. Sections were visualized after staining with 3,3-diaminobenzidine tetrahydrochloride (DAB, Sigma-Aldrich) and 0.02% $H_2O_2$ in Tris-buffered saline (pH 7.6). Slides were washed and were mounted in Entellan New (Merck Millipore, Billerica, MA, USA), coverslipped, and photographed under a light microscope (Axioplan II; Carl Zeiss) equipped with a DP26 camera (Olympus, Tokyo, Japan).

For the quantitative assessment of DAB intensity, we compared sample pairs (control vs. mutant) prepared (fixed, embedded into paraffin, cut into slices, and immunostained) on the same day and as per the same schedule and conditions, such as the duration of developing times. We used ImageJ software (National Institutes of Health, Bethesda, MD, USA) and the color deconvolution plugin for proper separation of the DAB color spectra, as previously described [97]. Briefly, the region of interest (ROI, see Fig 4C) was manually determined with the polygon or freehand tool, and the deconvoluted image was then analyzed pixel-by-pixel. The color threshold for the positive area was defined in the range of 61–125/255, and the ratio of the positive area to the total image area was calculated and presented as a percentage of the control sample ratio.

## Immunoblotting

Dissected tissues from P5-7 mice, such as the SG and whole brain, were lysed in homogenizing buffer [98] by sonicating in the presence of a protease inhibitor cocktail (Nacalai Tesque) and 1% Triton X-100. Total cell lysates were centrifuged at $800 \times g$ for 5 min at 4˚C, and the

supernatants were subjected to SDS-PAGE followed by immunoblotting for 2 h at 23°C using primary antibody. The bound primary antibodies were detected with secondary antibody-HRP conjugates using the ECL detection system.

For quantification, the relative expression levels of interested molecules were normalized to that of GAPDH, as previously described [96].

## ABR and DPOAE measurements

To assess hearing, ABR and distortion product otoacoustic emission (DPOAE) were measured in mice under anesthetization with a mixture of medetomidine, midazolam, and butorphanol (intraperitoneal, 0.3 mg/kg, 4.0 mg/kg, and 5.0 mg/kg, respectively) and on a heating pad. In mice, ABR and DPOAE are reportedly mature and saturated at 2 weeks and 2–4 weeks after birth, respectively [23]. Blinded data analysis was performed by two otologists.

ABR was measured in $Large^{myd}$, POMGnT1-KO, and $Dmd^{mdx}$ mice with their littermate control mice at the indicated ages, as described previously [96]. ABR waveforms using sound stimuli of clicks or tone bursts at 8 kHz, 16 kHz, 24 kHz, or 32 kHz were recorded for 12.8 ms at a sampling rate of 40,000 Hz using 50–5000 Hz band-pass filter settings, and ABR waveforms from 500 stimuli were averaged. ABR thresholds (dB SPL) were defined by decreasing the sound intensity by 5 dB intervals until the lowest sound intensity level, resulting in a recognizable ABR wave pattern (mainly judged by recognition of wave III), was achieved.

DPOAE was measured in $Large^{myd}$, POMGnT1-KO, $Dmd^{mdx}$ mice with their littermate controls at the indicated ages, as described previously [96]. DPOAE at frequency of 2f1–f2 were elicited using two primary tone stimuli, f1 and f2, with sound pressure levels of 65 and 55 dB SPL respectively, with f2/f1 = 1.20. DPOAE amplitudes (dB SPL) were measured at f2 frequencies of 4, 6, 8, 10, 12, 16, 18, and 20 kHz and plotted after substitution with noise floor amplitude.

## ABR studies in humans

The auditory function in Fukuyama CMD patients was evaluated using ABR. In order to avoid risks related to the sedation procedures in Fukuyama CMD patients, which is usually required for conventional ABR testing, we used the Integrity 500 System (Rion Co. Ltd., Tokyo, Japan) that can measure ABR in awake patients using a combination of in situ amplifier-electrodes and Kalman-weighted averaging [99]. Each participant was examined by otoscopy prior to the ABR testing, and participants with abnormal otoscopic findings were excluded from the study. The normal hearing in controls was verified by responses to pure tone audiometry or condition-oriented response audiometry at sound intensity of 40 dB in 1 kHz and 4 kHz. The ABR waveforms using click stimuli presented through insert-earphones (Intelligent Hearing Systems, Miami, FL) were recorded for 25.0 ms at a sampling rate of 27.5 Hz using 30–1500 Hz band-pass filter settings. Equivalent sweeps determined by Kalman weighted algorithm were spontaneously averaged and visualized as ABR waveforms, and the recording was continued until both wave I and V became recognizable and plateaued or the number of the equivalent sweeps became larger than 2000.

Obtained ABR waveforms with sound intensity of 40 dB SPL and 60 dB SPL were evaluated by two otologists. A non-recognizable wave pattern larger than 2000 equivalent sweeps was defined as no response. ABR thresholds were determined as minimal sound intensities at which recognizable wave V was obtained. After the thresholds were determined, the latency and amplitude of each wave were analyzed at 60 dB, since the waveform was more clearly determined at 60 dB than at 40 dB. The amplitudes and the latencies of wave I and V and the interpeak latency between wave I and V in Fukuyama CMD patients were analyzed and

compared with those in age- and sex-matched normal controls. An amplitude of a wave was defined as the voltage difference between the peak of the wave and the adjacent negative peak after the wave.

## TEM

Sample preparation and observation were conducted as reported previously [24]. Briefly, freshly dissected inner ear tissues were fixed in 2% paraformaldehyde, 2.5% glutaraldehyde in 0.1 M PB. OC epithelia were dissected in the same buffer and postfixed with 1% OsO4 in $H_2O$ for 1 h. For TEM analyses, samples were embedded in Spurr Low-Viscosity Embedding Media after post-fixation (Polysciences, Germany) and polymerized at 70˚C for 8 h. Ultra-thin sections (thickness ~70 nm) were cut using an ultramicrotome (EM-UC7; Leica Microsystems, Germany), placed on copper grids, and examined on Hitachi H-7100 electron microscope at 80 kV.

For the quantitative assessment of myelination, we classified abnormal axons into three categories in the transverse section at the OSL (see Fig 6A): "naked axons", "axons with disrupted myelin", and "axons with secondary changes (vacuoles and/or aggregates)". "Axons with abnormal myelination" was defined as the sum of "naked axons" and "axons with disrupted myelin." An "axon with myelination" (S5B Fig) was defined as a myelinated axon regardless of myelin morphology. We also measured the longest diameter of each myelinated axon (including axons with abnormal myelin, but excluding naked axons) in the transverse section at the OSL; their distribution is shown as a bar graph (x-axis: diameter, y-axis: number of axon) and boxplot.

## Statistical analysis

All data are presented as mean ± SE. Two groups were compared using the unpaired two-tailed Student's t-test, Kolmogorov–Smirnov test, or Mann-Whitney's U-test. For comparisons of more than two groups, one-way ANOVA or two-way ANOVA was performed, followed by Tukey's or Bonferroni's *post hoc* test of pairwise group differences. Statistical analyses were performed using Prism 7.0 software (GraphPad); a *P-value of* $< 0.05$ was considered statistically significant.

## Supporting information

**S1 Fig. DPOAE in *Large^myd/myd* and *POMGnT1*-KO mice. A,** DPOAE assessed using puretone bursts at 4, 6, 8, 10, 12, 16, 18, and 20 kHz in 5-week-old control (*Large^wt/wt*, $n = 6$), *Large^myd/wt* ($n = 4$), and *Large^myd/myd* mice ($n = 6$). DPOAE assessed using pure-tone bursts at 12 and 18 kHz in 2-, 5-, and 9-week-old control mice ($n = 6, 6$, and 4, respectively) and *Large^myd/myd* mice ($n = 5, 6$, and 4, respectively) was graphed. $^{****}P < 0.0001$ and $^{***}P = 0.0007$ (control vs. *Large^myd/myd*) using two-way ANOVA with Tukey's post-hoc test. **B,** DPOAE assessed with pure-tone bursts at 4, 6, 8, 10, 12, 16, 18, and 20 kHz in 6-week-old control (*wt/wt*, $n = 4$), heterozygous *POMGnT1*-KO (*ko/wt*, $n = 4$), and *POMGnT1*-KO (*ko/ko*, $n = 4$) mice. No significant difference was observed. (TIF)

**S2 Fig. Decreased CNPase levels in the SG/RC in *Large^myd/myd* mice.** Lysates were obtained from the spiral ganglion (SG)/Rosenthal's canal (RC) of the P5-7 control and *Large^myd/myd* mice. CNPase immunoblotting showed decreased levels of CNPase in the *Large^myd/myd* mice. Comparative loading of proteins was confirmed by immunoblotting of GAPDH. Statistical analysis was performed in pairs of the control ($n = 9$) and *Large^myd/myd* mice ($n = 7$), $^{**}P = 0.0015$ by Student's *t*-test. (TIF)

**S3 Fig. Normal hearing function and immunoreactivity of α-DG and MBP in *Dmd^{mdx/mdx}*
mice. A,** ABR with click and pure-tone bursts at 8, 16, 24, and 32 kHz and DPOAE with pure-
tone bursts at 8, 12, 16, and 20 kHz were performed in 12-week-old control (*Dmd^{wt/wt}*, *n* = 6)
and *Dmd^{mdx/mdx}* mice (*n* = 6). ABR latency of wave I at the click stimulation of 90 dB in con-
trol and *Dmd^{mdx/mdx}* mice (*n* = 4 and 6, respectively) were graphed. No significant differences
were observed in ABR and DPOAE analyses using two-way ANOVA with Bonferroni's post-
hoc test and in ABR latency of wave I analysis using Student's *t*-test (*P* = 0.1230). **B and C,**
Inner ears of 12-week-old control and *Dmd^{mdx/mdx}* mice were fixed for immunostaining of gly-
cosylated α-DG [α-DG(gly)] and core α-DG [α-DG(core)] proteins (**B**; *n* = 4 and 6, respec-
tively) and MBP (**C**; *n* = 5). Statistical analysis of immunoreactivity was performed and
graphed. No significant difference was observed between control and *Dmd^{mdx/mdx}* mice using
Student's *t*-test (*P* = 0.2039 in α-DG(gly), *P* = 0.1597 in α-DG(core), and *P* = 0.4902 in MBP).
Scale bars: 50 μm.
(TIF)

**S4 Fig. Quantification of abnormal myelination and axons at the OSL in *Large^{myd/myd}* and
*POMGnT1*-KO mice.** Inner ears of control and *Large^{myd/myd}* mice (**A, B**) and control and
*POMGnT1*-KO mice (**C, D**) were fixed for transmission electron microscopy (TEM). TEM
images at the osseous spiral lamina (OSL) were obtained. Each percentage was obtained by
analyzing 50 axons. Statistical analyses were conducted using the Student's *t*-test unless indi-
cated. **A,** Graph showing the percentages of naked axons (****P* < 0.0001) and axons with dis-
rupted myelin (***P* = 0.0016) in 6-week-old control (*n* = 5) and *Large^{myd/myd}* (*n* = 6) mice. **B,**
Graph showing the percentage of axons with secondary changes in 2-week-old control and
*Large^{myd/myd}* mice (*n* = 3, *P* = 0.5614), 6-week-old control (*n* = 5) and *Large^{myd/myd}* (*n* = 6)
mice (**P* = 0.0100), and 10-week-old control (*n* = 5) and *Large^{myd/myd}* (*n* = 5) mice
(***P* = 0.0089). **P* = 0.0398 (*Large^{myd/myd}* mice at 2 vs. 10 weeks) using one-way ANOVA with
Tukey's post-hoc test. **C–D,** Graph showing the percentages of naked axons (***P* = 0.0024),
axons with disrupted myelin (**P* = 0.0278), and axons with secondary changes (*P* = 0.6918) in
control and *POMGnT1*-KO mice (*n* = 3).
(TIF)

**S5 Fig. Myelination at the OSL and proximal to the glial dome in *Large^{myd/myd}* mice.** Inner
ears of 6- and 10-week-old control and *Large^{myd/myd}* mice (**A**) and 5-week-old control and *Lar-
ge^{myd/myd}* mice (**B**) were fixed for transmission electron microscopy (TEM). TEM images at the
osseous spiral lamina (OSL, **A**) and proximal to the glial dome (GD) (**B,** at the region indicated
by the asterisk in Fig 4A) were obtained. Cochleae of 8-week-old control and *Large^{myd/myd}* mice
(**C**) were fixed for MBP immunostaining. **A,** The longest diameter of each myelinated axon in the
transverse section at the OSL in *Large^{myd/myd}* mice at 6 weeks (*n* = 100) and 10 weeks (*n* = 74)
were statistically analyzed (3 cochleae) using the Kolmogorov–Smirnov test. No significant differ-
ence was observed (*P* = 0.3043). **B,** The percentage of axons with myelination and of axons with
abnormal myelinations in control and *Large^{myd/myd}* mice were calculated per x 5000-field, and
statistically analyzed (total 10 fields of each obtained from 3 control and 5 *Large^{myd/myd}* mice)
using the Student's *t*-test. Lower panels represent magnified images of the areas indicated by the
squares in the upper panels. No significant difference was observed in axons with myelination
(*P* = 0.1657), but a significant difference was observed in axons with abnormal myelination
(*****P* < 0.0001). Arrowheads indicate abnormal myelination. Scale bars: 1 μm. **C,** Immunostain-
ing was performed using an MBP antibody, and statistical analyses were conducted between con-
trol and *Large^{myd/myd}* mice (*n* = 3). No significant difference was observed by Student's *t*-test
(*P* = 0.1984). RC: Rosenthal's canal. Scale bars: 100 μm.
(TIF)

**S6 Fig. Decreased MBP levels in the brain in *Large^myd/myd* mice. A,** Brain sections of eight-week-old control and *Large^myd/myd* mice at the level of the corpus callosum were obtained for immunostaining for MBP (Scale bars: 200 μm). The panels on the right are magnified images of the corpus callosum indicated by the squares in the panels on the left (Scale bars: 100 μm). Decreased immunoreactivity of MBP was observed in *Large^myd/myd* mice. The results are presented as the mean of at least three experiments. **B,** Whole-brain lysates of P7 control and *Large^myd/myd* mice (*n* = 5) were obtained for MBP immunoblotting. Decreased expression levels of MBP were observed in *Large^myd/myd* mice compared with those in control mice. Comparative loading of proteins was confirmed by immunoblotting of GAPDH. $^*P$ = 0.0121 by Student's *t*-test.
(TIF)

**S7 Fig. No apparent morphological anomaly in the cochlea in *Large^myd/myd* mice in TEM images.** Organs of Corti of 10-week-old controls and *Large^myd/myd* mice were fixed for transmission electron microscopy (TEM) (**A and B**). TEM images at the apical connective spaces between outer hair cells (OHCs) and supporting cells (SCs) (**A**, indicated by the rectangle) and between OHCs and underlying basal lamina (BL) (**B**, indicated by the rectangle). No apparent difference was observed between control and *Large^myd/myd* mice. Representative results from three independent experiments are shown (*n* = 3). Scale bars: 500 nm.
(TIF)

**S1 Table. ABR analysis of Fukuyama CMD patients.** Latency of wave I (latency I) and wave V (latency V), interpeak latency between wave I and V (interpeak I-V), and amplitude of wave I (amplitude I) in nine Fukuyama CMD patients analyzed in the present study are shown. Severity is classified based on the physical activity: mild, able to crawl; moderate, able to sit; severe, unable to control head position. CC, cerebellar cyst; ID, intellectual disability.
(DOCX)

**S2 Table. ABR analysis of healthy volunteers (controls) evaluated in the present study.** Latency of wave I (latency I) and wave V (latency V), interpeak latency between wave I and V (interpeak I-V), and amplitude of wave I (amplitude I) in controls are shown.
(DOCX)

**S3 Table. Comparison between all Fukuyama CMD patients and controls.** Participant number, sex, mean age, and ABR data were compared between all Fukuyama CMD patients (total) and controls.
(DOCX)

**S4 Table. Comparison between Fukuyama CMD patients with homozygous mutations and controls.** Participant number, sex, mean age, and ABR data were compared between Fukuyama CMD patients with homozygous (homo) mutations and controls.
(DOCX)

**S5 Table. Comparison between Fukuyama CMD patients with heterozygous mutations and controls.** Participant number, sex, mean age, and ABR data were compared between Fukuyama CMD patients with heterozygous (hetero) mutations and controls.
(DOCX)

**S1 Data. Raw data of Fig 2A.**
(XLSX)

**S2 Data. Raw data of Fig 2B.**
(XLSX)

**S3 Data. Raw data of Fig 3.**
(XLSX)

**S4 Data. Raw data of Figs 4–7.**
(XLSX)

**S5 Data. Raw data of S1 Fig.**
(XLSX)

**S6 Data. Raw data of S2 Fig.**
(XLSX)

**S7 Data. Raw data of S3 Fig.**
(XLSX)

**S8 Data. Raw data of S4–S6 Figs.**
(XLSX)

## Acknowledgments

We thank Rion Co. Ltd., Tokyo, Japan for free offer to use the Integrity 500 System, which can measure ABR in awake patients. We also thank Ms Mai Kondo, Division of Molecular Brain Science, Kobe University Graduate School of Medicine, for the technical assistance.

## Author Contributions

**Conceptualization:** Takehiko Ueyama.

**Data curation:** Hirofumi Sakaguchi, Naoaki Saito, Takehiko Ueyama.

**Formal analysis:** Hirofumi Sakaguchi, Takehiko Ueyama.

**Funding acquisition:** Hirofumi Sakaguchi, Naoaki Saito, Takehiko Ueyama.

**Investigation:** Shigefumi Morioka, Hirofumi Sakaguchi, Hiroaki Mohri, Mariko Taniguchi-Ikeda, Toshiaki Suzuki, Takehiko Ueyama.

**Methodology:** Takehiko Ueyama.

**Project administration:** Takehiko Ueyama.

**Resources:** Motoi Kanagawa, Yuko Miyagoe-Suzuki, Tatsushi Toda.

**Supervision:** Hirofumi Sakaguchi, Motoi Kanagawa, Tatsushi Toda, Naoaki Saito, Takehiko Ueyama.

**Validation:** Hirofumi Sakaguchi, Takehiko Ueyama.

**Visualization:** Hirofumi Sakaguchi, Takehiko Ueyama.

**Writing – original draft:** Shigefumi Morioka, Hirofumi Sakaguchi, Motoi Kanagawa, Takehiko Ueyama.

**Writing – review & editing:** Shigefumi Morioka, Hirofumi Sakaguchi, Hiroaki Mohri, Mariko Taniguchi-Ikeda, Motoi Kanagawa, Tatsushi Toda, Naoaki Saito, Takehiko Ueyama.

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
