## [Decision Letter · Decision Letter 0]

31 Dec 2019

Dear Dr Ueyama,

Thank you very much for submitting your Research Article entitled 'Impaired α-dystroglycan sugar chains at the peripheral cochlear nerve cause congenital hearing loss with dysmyelination' to PLOS Genetics. Your manuscript was fully evaluated at the editorial level and by independent peer reviewers. The reviewers appreciated the attention to an important problem, but raised some major and minor concerns about the current manuscript. Based on the reviews, we will not be able to accept this version of the manuscript, but we would be willing to review again a much-revised version. We cannot, of course, promise publication at that time.

Should you decide to revise the manuscript for further consideration here, your revisions should address the specific points made by each reviewer. Reviewer #1 has raised some significant concerns each of which requires your attention. We will also require a detailed list of your responses to the review comments and a description of the changes you have made in the manuscript.

If you decide to revise the manuscript for further consideration at PLOS Genetics, please aim to resubmit within the next 60 days, unless it will take extra time to address the concerns of the reviewers, in which case we would appreciate an expected resubmission date by email to plosgenetics@plos.org.

[LINK]

We are sorry that we cannot be more positive about your manuscript at this stage. Please do not hesitate to contact us if you have any concerns or questions.

Yours sincerely,

Thomas B. Friedman

Guest Editor

PLOS Genetics

Hua Tang

Section Editor: Natural Variation

PLOS Genetics

Reviewer's Responses to Questions

**Comments to the Authors:**

Reviewer #1: This is a solid descriptive study of hearing loss in two mouse models of muscular dystrophy. Muscular dystrophy is caused by degeneration of skeletal muscles, often due to changes in the dystrophin-dystroglycan complex, which mediates attachments between cells and the extracellular matrix. Because this complex is present in other tissues, some forms of MD may include additional phenotypes, such as sensorineural hearing loss. The origin and nature of MD associated hearing loss is not well defined. This study provides new evidence that hearing may be affected in some forms of MD, possibly due to changes in myelination. Specifically, the data show that 1) ABR and DPOAE thresholds are elevated in Large(myd/myd) mutant mice, with decreased ABR wave 1 amplitudes and increased wave 1 latencies; 2) ABR but not DPOAE thresholds are modestly elevated in POMGnT1-KO mice; 3) alpha-DG and laminin alpha2 levels are diminished in the myd/myd and POMGnT1-KO mice, as shown by immunostaining and Western blot; 4) myelination is affected, with decreased MBP and CNPase levels and structurally abnormal myelin shown by EM; 5) Wave 1 is also delayed in patients with Fukuyama congenital MD. Thus, the authors conclude that “Impaired alpha-dystroglycan sugar chains at the peripheral cochlear nerve cause congenital hearing loss with dysmyelination.”

Overall, the study presents interesting new data highlighting the possibility of hearing loss in some (but not all) forms of MD. The data convincingly demonstrate that hearing is affected in two mouse models, both of which show associated changes in the levels of glycosylated alphya-dystroglycan and laminin in the region of the cochlea where SGN peripheral processes are myelinated. Further, the authors show changes in myelination that are stronger distal to the glial dome than proximal. My major criticism is that the data do not firmly link the observed cellular changes to the auditory phenotype. Specifically, although the data do support the conclusion that both mouse strains exhibit congenital hearing loss with dysmyelination, the data do not prove that the changes in the sugar changes in the peripheral nerve are responsible for all of the reported phenotypes (i.e. elevated thresholds, abnormal DPOAEs). Additionally, the dysmyelination phenotype needs to be quantified better.

Major points:

1. The title is not accurate. The studies convincingly show that alpha-dystroglycan and laminin protein levels are decreased at the peripheral cochlear nerve and that the mice do not hear well, but the experiments do not directly link these two observations. In fact, it is striking that the hearing phenotypes differ in the two mouse models: DPOAEs are essentially absent in the myd/myd mice but normal in the POMGnT1-KO mice. Since there is a strong DPOAE phenotype in the myd/myd mice, it is impossible to interpret the decreased wave 1 amplitude and increased latency, as these changes could be secondary to abnormal cochlear function. The POMGnT1-KO mice show only a transient change in wave I latency though thresholds are elevated at all ages. Since the abnormal myelination is shown at 10 weeks in that mutant, a time when latencies are recovered, it seems like the change in myelination cannot be responsible for the change in latency. Also, since both mice show changes in alpha-dystroglycan glycosylation and dysmyelination in the peripheral cochlear nerve yet only one strain shows a DPOAE phenotype, these cellular phenotypes are unlikely to be the full explanation for the observed hearing loss. Other interpretations need to be considered.

2. The authors demonstrate changes in alpha-dystroglycan and laminin using both immunohistochemistry and Western blot. However, it is hard for me to interpret the immunohistochemistry since it is an enzymatic detection using DAB substrate, which is not a quantitatively reliable assay. The authors should note whether the control and mutant tissues were processed on the same slides and developed for the same amount of time. This information was not provided in the Methods section.

3. The dysmyelination phenotype is striking but needs to be better quantified. I do not understand what is meant by “dispersion of the longest diameter in the myelinated axon” (p. 10, 212-210). This made it very hard for me to assess the data shown in Fig. 6. Please explain exactly what was measured and how. Likewise, I do not understand the definition of “normal myelinated axons” or how that was assessed objectively. Since the effect on myelination in both mouse lines is a key finding, these data should be better quantified overall. For instance, Fig 6 illustrates naked axons, disrupted myelin, vacuolated axons, and aggregated axons. The authors should quantify the incidence of each of these pathologies, blind to genotype, in mutants and littermate controls.

4. The authors note that “None of the FCMD patients analyzed had complaint of deafness” (line 236, page 11), but go on to show that wave 1 latencies were increased. The authors should discuss this point in the Discussion, as it has an impact on the clinical value of the information presented. For instance, please discuss how increased latencies might affect hearing and whether there are assays that might reveal a deficit that is otherwise missed by both the audiologists and the patients.

5. Since the primary cellular phenotype seems to be a change in myelination, there should also be some discussion of how the hearing phenotype compares to what is observed in other mouse models with impaired myelination.

6. For the patient analysis, the authors should report auditory thresholds so that the effect on latencies can be properly interpreted. I am also curious whether patients with “mild” motor phenotypes also had “mild” effects on latency.

Minor points:

1. I expect this paper will be interesting to hearing researchers who are unlikely to be familiar with the various forms of muscular dystrophy. The introduction needs to be improved to make it easier for non-experts to follow the differences between the three mouse models presented as well as the human form of MD that is assessed. This could be achieved by reworking the introduction and perhaps adding a figure introducing the various enzymes related to the mouse and human mutants that are ultimately examined.

2. I do not think it is appropriate to use a phrase such as “To prove our hypothesis.” This wording suggests that the authors wanted to be right. It is more appropriate to say “To test our hypothesis.” This may seem minor but I think it is important to demonstrate a value for unbiased and objective approaches in official scientific publications.

3. The paper should be proofread and edited for improved clarity. For instance, I believe “torn bursts” is meant to be “tone bursts” (line 420, p. 20) and “spinal ganglion” should be “spiral ganglion” (line 822, p. 36).

Reviewer #2: In the manuscript by Morioka et al., the authors catalog numerous phenotypes in mainly two mouse models (Large and Pomgnt1 deletion lines) for muscular dystrophy. Both mouse models show hearing impairment with some aspects being progressive. Importantly, the authors zero in on various Schwann cell phenotypic defects in their models, including defects in myelination, alpha-Dystroglycan expression, laminin expression and others. The data from the mouse models is correlated with auditory defects reported in patients with muscular dystrophy. Thus, with regard to muscular dystrophy, the authors draw a reasonable link between their mouse models and a patient population. In their analyses, the authors performed a battery of quantitative comparisons (between mutants and WT) and the sample power and statistical analyses throughout the manuscript are excellent. The experiments are reported in logical sequence and appropriate control experiments are provided.

One complaint is that the title of the manuscript suggests the authors have discovered a causal link between the loss of alpha-dystroglycan and hearing loss. This is not at all accomplished and so a more conservative title is strongly recommended.

The Discussion should provide more information on important future directions, like an investigation of the temporal and spatial expression patterns of Large and Pomgnt1. There should also be a more thorough discussion of possible defects in efferent fibers and OHC function, which could underlie the DPOAE defects reported. Also, have the Schwann cells started to die off at some point? Are they poorly differentiated? These questions should also be addressed in the Discussion.

A few minor issues:

The authors use excessive acronyms and these add a lot of visual clutter to the narrative. I suggest choosing only two or three terms to reduce to acronyms, and to otherwise spell out everything else.

The statement in the last two sentences of the 2nd paragraph (Discussion section, line 278-281) needs references to back up the hypothesis.

Figure 4D and S2B: Can authors show quantitative data for this experiment?

Line 399: Is this reference to figure 4C accurate?

Figure S6: Is this figure missing quantitative data? The Figure legend says that the results are presented as mean of at least 3 representative experiments.

Figure S6: The cartoon has some thick lines extending from HCs. What are these?

**Have all data underlying the figures and results presented in the manuscript been provided?**

Reviewer #1: Yes

Reviewer #2: No: It does not appear that all numerical data that goes into graphs and statistics has been provided.

PLOS authors have the option to publish the peer review history of their article (what does this mean?). If published, this will include your full peer review and any attached files.

Reviewer #1: No

Reviewer #2: No

---

## [Decision Letter · Decision Letter 1]

28 Apr 2020

Dear Dr Ueyama,

Thank you very much for submitting your Research Article entitled 'Congenital hearing impairment associated with peripheral cochlear nerve dysmyelination in glycosylation-deficient muscular dystrophy' to PLOS Genetics. Your manuscript was fully evaluated at the editorial level and by independent peer reviewers. The reviewers appreciated the attention to an important topic but identified some aspects of the manuscript that should be improved.

We therefore ask you to modify the manuscript according to the review recommendations before we can consider your manuscript for acceptance. Your revisions should address the specific points made by each reviewer.

[LINK]

Yours sincerely,

Thomas B. Friedman

Guest Editor

PLOS Genetics

Hua Tang

Section Editor: Natural Variation

PLOS Genetics

Reviewer's Responses to Questions

**Comments to the Authors:**

Reviewer #2: The authors put forth great effort to address the concerns that came from the first submission. The manuscript is well-suited for publication. One note -- during the revisions, the authors apparently had some survival issues in their POMGnT1-KO mice, which left them with limited sample numbers for quantifying Western blots. They included what they had in Supplementary Figures 2A and 2C. However, the small sample size precluded statistical significance and so statistical data are not shown. Given the small sample size, my opinion is that these data are preliminary and not suitable for publication. I suggest the authors remove Supplementary Figures 2A and C and note the trends and limited sample numbers in the Results section.

**Have all data underlying the figures and results presented in the manuscript been provided?**

Reviewer #2: None

PLOS authors have the option to publish the peer review history of their article (what does this mean?). If published, this will include your full peer review and any attached files.

Reviewer #2: No

---

## [Editor Report · Decision Letter 2]

4 May 2020

Dear Dr Ueyama,

We are pleased to inform you that your manuscript entitled "Congenital hearing impairment associated with peripheral cochlear nerve dysmyelination in glycosylation-deficient muscular dystrophy" has been editorially accepted for publication in PLOS Genetics. Congratulations!

Yours sincerely,

Thomas B. Friedman

Guest Editor

PLOS Genetics

Hua Tang

Section Editor: Natural Variation

PLOS Genetics

Comments from the reviewers (if applicable):

**Data Deposition**

http://datadryad.org/submit?journalID=pgenetics&manu=PGENETICS-D-19-01933R2

**Press Queries**

---

## [Editor Report · Acceptance letter]

20 May 2020

PGENETICS-D-19-01933R2 

Congenital hearing impairment associated with peripheral cochlear nerve dysmyelination in glycosylation-deficient muscular dystrophy 

Dear Dr Ueyama, 

We are pleased to inform you that your manuscript entitled "Congenital hearing impairment associated with peripheral cochlear nerve dysmyelination in glycosylation-deficient muscular dystrophy" has been formally accepted for publication in PLOS Genetics! Your manuscript is now with our production department and you will be notified of the publication date in due course.

With kind regards,

Jason Norris

PLOS Genetics

On behalf of:
